# Metabolic and genetic risk factors associated with pre-diabetes and type 2 diabetes in Thai healthcare employees: A long-term study from the Siriraj Health (SIH) cohort study

Pichanun Mongkolsucharitkul[1], Apinya Surawit[1], Thamonwan Manosan[1], Suphawan Ophakas[1], Sophida Suta[1], Bonggochpass Pinsawas[1], Tanyaporn Pongkunakorn[1], Sureeporn Pumeiam[1], Winai Ratanasuwan[2], Mayuree Homsanit[2], Keerati Charoencholvanich[3], Yuthana Udomphorn[4], Bhoom Suktitipat[5], Korapat Mayurasakorn[1]*

1 Siriraj Population Health and Nutrition Research Group, Research Department, Faculty of Medicine Siriraj Hospital, Mahidol University, Bangkok, Thailand, 2 Department of Preventive and Social Medicine, Faculty of Medicine Siriraj Hospital, Mahidol University, Bangkok, Thailand, 3 Department of Orthopedic Surgery, Faculty of Medicine Siriraj Hospital, Mahidol University, Bangkok, Thailand, 4 Department of Anesthesiology, Faculty of Medicine Siriraj Hospital, Mahidol University, Bangkok, Thailand, 5 Department of Biochemistry, Faculty of Medicine Siriraj Hospital, Mahidol University, Bangkok, Thailand

☯ These authors contributed equally to this work.
* korapat.may@mahidol.ac.th

## Abstract

### Background

The study of non-communicable diseases (NCDs) in a developing country like Thailand has rarely been conducted in long-term cohorts, especially among the working-age population. We aim to assess the prevalence and incidence of risk factors and their associations underlying NCDs, especially type-2 diabetes mellitus (T2DM) among healthcare workers enrolled in the Siriraj Health (SIH) study cohort.

### Methods

The SIH study was designed as a longitudinal cohort and conducted at Siriraj hospital, Thailand. A total of 5,011 participants (77% women) were recruited and follow-up. Physical examinations, blood biochemical analyses, family history assessments, behavior evaluations, and genetics factors were assessed.

### Results

The average age was 35.44±8.24 years and 51% of participants were overweight and obese. We observed that men were more likely to have a prevalence of T2DM and dyslipidemia (DLP) compared to women. Aging was significantly associated with pre-diabetes and T2DM (*P*<0.001). Additionally, aging, metabolic syndrome, and elevated triglycerides were associated with the development of pre-diabetes and T2DM. The minor T allele of the rs7903146(C/T) and rs4506565 (A/T) were associated with a high risk of developing pre-

**Data Availability Statement:** The study is a part of the Siriraj Health (SIH) Cohort Study. The authors

do not have the legal authority to distribute the data. However, all interested researchers can follow the respective rules and protocols of data sharing and scientific collaboration, just as the current authors have. The data are accessible upon request from the Siriraj Health study committee via email: sihealthstudy@mahidol.edu.

**Funding:** This work was supported by the Faculty of Medicine Siriraj Hospital, Mahidol University (R016034006). Additional funding support including infrastructure, staff and utilities, was provided by Faculty of Medicine Siriraj Hospital, Mahidol University. The funder had no role in the study design, data collection and analysis, decision to publish, and preparation of the manuscript.

**Competing interests:** The authors have declared that no competing interests exist.

diabetes with odds ratios of 2.74 (95% confidence interval [CI]: 0.32–23.3) and 2.71 (95% CI: 0.32–23.07), respectively; however, these associations were statistically insignificant ($P > 0.05$).

## Conclusion

The findings of the SIH study provide a comprehensive understanding of the health status, risk factors, and genetic factors related to T2DM in a specific working population and highlight areas for further research and intervention to address the growing burden of T2DM and NCDs.

## Introduction

Non-communicable diseases (NCDs) represent a global health crisis, driven by factors such as rapid urbanization, unhealthy diets, physical inactivity, and an aging population [1–4]. Metabolic risk factors, including high blood glucose, dyslipidemia (DLP), and obesity, contribute to 41 million NCD-related deaths annually, primarily among those aged 30 to 69 years [5–7]. Thailand, an upper middle-income country (UMIC), has seen a substantial rise in its older adult population from 10.7% in 2007 to 19.2% in 2020, with a projected surge to 23% by 2030, aligning with the broader trends observed in the Southeastern Asia region (12.2 to 16.7% in 2020; and 15.7 to 20% in 2030, respectively) [8, 9]. In 2018, half of Thailand's population resided in urban areas, and the United Nations predicts that 68% of the global population will be urban dwellers by 2050 [10]. This demographic shift poses substantial challenges to healthcare and sustainable development. Recent data from the Thailand National Health Examination Survey showed that nearly 30% of Thai adults over 40 exhibit dysglycemia, mainly concentrated in urban areas [11]. Obesity, hypertension (HTN), and type-2 diabetes mellitus (T2DM), NCD risk factors, have surged in Thailand in the last decade [12–14].

Promoting healthy diets and physical activity is vital to mitigate NCDs across the lifespan [15, 16]. Maintaining good glycemic control can prevent microvascular complications [17]. Addressing modifiable risk factors early and empowering younger populations with relevant knowledge and interventions are crucial steps. Urbanization and mobility bring both challenges and opportunities for research. A quarter of Thai Open University students moved from rural to urban areas from 2005 to 2009, impacting their health behaviours [18]. The Electricity Generating Authority of Thailand (EGAT) cohort study found diverse socioeconomic backgrounds, a wealthier profile, higher male participation in smoking and alcohol, and an upper-class healthcare scheme [19]. Longitudinal urban cohort studies focused on NCDs are essential.

Therefore, we established a long-term cohort database at a Thai urban medical university, implementing rigorous data quality control. A biobank has been established to store specimens for the assessment of genetic factors pertinent to NCD development, especially T2DM. The TCF7L2 gene plays a crucial role in pancreatic β-cell proliferation and insulin secretion regulation. Changes in this pathway can lead to T2DM [20]. A common single nucleotide polymorphism (SNP) in the TCF7L2 gene region is associated with T2DM [21]. In Asian populations, including Japanese [22], Thai [23], and Chinese [24], variants of the TCF7L2 gene, such as rs7903146, rs11196205, and rs12255372, have been identified as significant genetic risk factors for T2DM. These findings highlight the genetic heterogeneity of T2DM across different ethnic groups and underscore the importance of understanding population-specific genetic

determinants of the disease. Our study aims to explore NCD risk factors, biomarker relationships, and develop a T2DM risk prediction model while investigating the association between T2DM and genetic variants of the TCF7L2 gene.

## Methods

### Study population

The Siriraj Health (SIH) study, conducted at Siriraj Hospital, Mahidol University, Bangkok, Thailand (Fig 1), is a comprehensive longitudinal cohort comprising diverse healthcare professionals (doctors, nurses, pharmacists, medical technicians etc.), support staff (drivers, engineers, security officers, clerks etc.), and academic personnel (lecturers, researchers, research assistants etc.). As of November 2023, there were 20,967 individuals working in this university hospital and approximately 80% undergo annual health check-up. The inclusion criteria of the study were the Siriraj Hospital personnel who attended an annual health screening surveillance program. The exclusion criteria of the study were individuals who withheld treatment information, unable to consistently follow-up, unable to participate in this cohort in the next 2 years, or presence of contraindications for blood sampling, such as blood clotting. SIH provides a platform for in-depth studies of extensive, long-term data, and biological specimens from Siriraj Hospital personnel based on an annual health screening surveillance program. This study estimated the sample size at 5,000 individuals based on NCD prevalence with a 95% confidence interval (CI) and standardized for age and sex according to the total population of the study [11]. Between September 2017 and December 2019, 4,518 participants aged 18 to 55

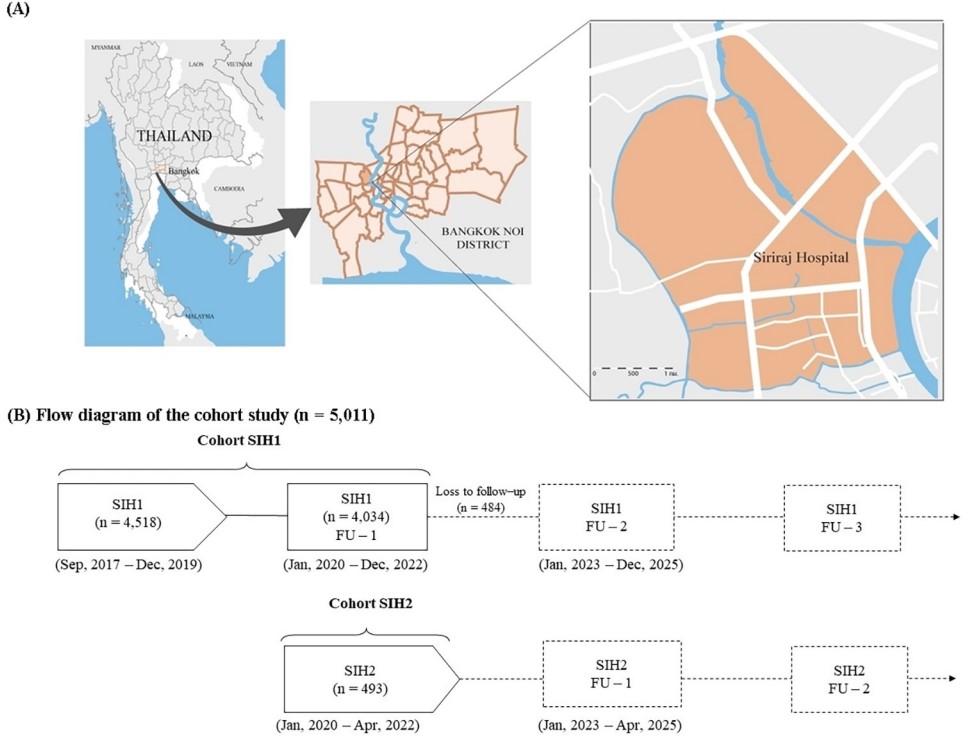

**Fig 1. The Siriraj Health (SIH) study.** (A) Geographic distribution of Siriraj Health study (marked area) in the center of Bangkok, Thailand; (B) Flow diagram of participants enrolled in the Siriraj Health cohort study; FU, Follow-up; SIH, Siriraj Health. Reprinted under a CC BY license, with permission from Ms. Apinya Surawit, original copyright [2024].

years were enrolled in phase 1 (SIH1). Beginning in January 2020, the second phase of the Siriraj Health (SIH2) study included participants aged over 18 years with no upper age limit. However, due to the COVID-19 pandemic from 2020 to 2022, only 493 participants were enrolled. A total of 5,011 participants are illustrated in Fig 1. The follow-up data from health check-up in 2020 of SIH1's participants were retrieved from electronic medical records in the hospital database. Out of the 4,518 people enrolled in the first cohort (SIH1), 484 participants (10.7%) were lost to follow-up in 2020 due to death, changes of a workplace, and loss of contact.

## Data collection

During the annual health check-up, Siriraj's personnel were provided with initial information about the SIH cohort study. Written informed consent was obtained from all participants before enrollment and specimen collection. The SIH study was conducted in accordance with the principles established by the Declaration of Helsinki and was approved by the Ethics Committee of the Human Research Protection Unit, Faculty of Medicine Siriraj Hospital, Mahidol University board (COA no. Si 647/2016). We provided instructions on the study's standard operating procedures (SOPs) to a team of ten well-trained research staff members involved in the SIH study. These staff members were responsible for securing informed consent, conducting face-to-face questionnaire interviews, and collecting biological specimens. This collection took place within a well-equipped laboratory equipped with a standard biobank for preserving deoxyribonucleic acid (DNA) and plasma samples. A decoded identification number was generated for each participant, which was used to label all biological specimens and subject data. The management of study data was entrusted to a bioinformatician utilizing the R program (version 4.1.3, Revolution Analytics, Dallas, TX, USA) and Research Electronic Data Capture (REDCap, version 12.0.13, Vanderbilt, TN, USA) [25]. The computer servers were located at the Siriraj Informatics and Data Innovation Center (SiData+), part of the Faculty of Medicine Siriraj Hospital, Mahidol University.

## Physical examinations

Trained nurses from the Department of Preventive and Social Medicine, Faculty of Medicine Siriraj Hospital, Mahidol University, interacted with participants using established standard procedures. Data measurements included weight, height, waist circumference (WC), and blood pressure (BP). Weight measurements were obtained to the nearest 0.1 kg (Tanita BWB-800, Tanita Corporation, Tokyo, Japan), while standing height was measured to the closest 0.1 cm. WC was determined using a flexible, non-stretchable plastic tape positioned across the midpoint between the lowest rib and the upper lateral border of the right iliac crest. Systolic blood pressure (SBP) and diastolic blood pressure (DBP) measurements were conducted using a digital blood pressure monitor (HEM-907, Omron Corporation, Tokyo, Japan).

## Laboratory measurements

A 34 ml venous blood sample was collected after a 12-hour fast, and 30 ml of urine was collected from each participant. To prevent the occurrence of hypoglycemic events, all blood samples were collected between 07:00 and 09:00 A.M. Blood samples were distributed into four anticoagulation tubes for complete blood count (CBC), glycated hemoglobin (HbA1c) analysis, plasma separation, and DNA extraction. A separate sodium fluoride tube inhibited glycolysis for fasting blood glucose (FBG) analysis, while a lithium heparin tube facilitated total cholesterol (TC), triglyceride (TG), high-density lipoprotein cholesterol (HDL-C), low-density lipoprotein cholesterol (LDL-C), creatinine (Cr), and estimated glomerular filtration rate

(eGFR) assessment. Urine samples were analyzed for spot urine albumin-to-creatinine ratio (MAU/Cr). Blood samples were centrifuged at 3,000 rounds per minute for 10 minutes at 4°C, and plasma samples were aliquoted into cryotubes (Thermo Fisher Scientific, Jiangsu, China). Biochemical assays were performed using automated systems with various methods by an accredited clinical laboratory (Siriraj hospital, Thailand) as shown in S1 Table. We used the criteria of metabolic syndrome (MetS) in adults by the IDF definition with South Asian ethnic group [26]. According to the IDF definition, MetS is present if three or more of the following five criteria are met: WC $\geq$ 90 cm (men) or $\geq$ 80 cm (women), BP $\geq$ 130/85 mmHg, TG level $\geq$ 150 mg/dL, HDL-C level < 40 mg/dL (men) or < 50 mg/dL (women) and FBG $\geq$ 100 mg/dL.

## DNA extraction and genotyping

Human DNA was extracted from 4 ml of whole venous blood using the Chemagic 360 automated platform and Chemagic DNA blood kits (CMG-1074, PerkinElmer, Baesweiler, Germany), following the manufacturer's instructions. DNA concentration and a purity ratio (within the acceptable in range 1.8–1.9) were determined using FLUOstar Omega (software version 5.5 R4, BMG LABTECH, Ortenberg, Germany). DNA samples were stored at -80°C in the Siriraj Biobank. Genotyping involved the analysis of 3,960 DNA samples for SNPs using Infinium Asian Screening Array (ASA, Illumina, San Diego, CA, USA). The basic microarray technical data for ASA were downloaded from Illumina official website (http://www.illumina.com). An extensive analysis of 659,184 SNPs was analyzed from each individual. Before conducting the genotyping analysis, we assessed the quality call rate for both samples (97% cutoff) and SNPs (90% cutoff), excluding those with call rates below the specified thresholds. Additionally, a comparison was made between the reported sex based on demographic data and the sex predicted from genotypes. Genotyping analysis was performed using the PLINK program, version 1.9 (S1 File).

## Self-report underlying diseases and family histories

Participants completed questionnaires about underlying diseases (T2DM, HTN, DLP, ischemic heart, stroke, and gout) and family history (heart failure, cancer, and stroke). HTN was defined as a SBP and DBP above the threshold (120/80 mmHg) or the current use of antihypertensive medication [27]. The diagnostic HbA1c criteria for non-diabetes (non-DM) was < 5.7%, pre-diabetes (pre-DM) 5.7–6.4%, and diabetes mellitus (DM) > 6.4%, or the current use of hypoglycemic medication [28]. Individuals who reported that they did not have DM but exhibited abnormal biochemical measurements within range of pre-DM were classified as unaware pre-DM [29] (S1 Fig).

## Behavioral risk factors

Participants completed questionnaires about exercise, smoking, and alcohol consumption. Regular exercise was defined as an exercise at least once a week. Smoking was defined as either a former smoker or a current smoker. Alcohol drinking habits were defined as individuals who consumed alcohol at least once a month in the past year.

## Statistical analysis

All continuous variables in the demographic data were expressed as the mean ± standard deviation (SD), whereas binary variables were expressed as numbers and percentages. Self-report data were expressed as percentage with 95% CI. All statistical analyses were conducted using

STATA version 14 (STATA Corp., Texas). The unadjusted cross-sectional associations of each risk factor at baseline with non-DM, pre-DM, and T2DM were examined using Chi-squared ($\chi^2$) tests. Adjusted odds ratios (OR) and 95% CI for cross-sectional associations at baseline were calculated using multivariable binary logistic regression models. Separate models were used for pre-DM versus non-DM and T2DM versus non-DM. To assess the longitudinal association between risk factors and progression to pre-DM or T2DM at baseline, the sample was restricted to individuals without T2DM at baseline. Multinomial logistic regression models were then used to calculate adjusted relative risk ratios (RRR) and 95% CI for T2DM-related outcomes (non-DM, pre-DM, T2DM) at follow-up. The average predicted probabilities of being classified as having normal blood glucose, pre-DM, or T2DM, given specific characteristics (each risk factor of interest for longitudinal analyses), were obtained using the Stata margins command. In this study, we focused on seven SNPs of the TCF7L2 gene, including rs7903146 (C/T), rs12255372 (G/T), rs7917983 (C/T), rs4506565 (A/T), rs4132670 (C/T), rs12243326 (C/T), and rs290487 (C/T). Genotype frequencies were tested for Hardy-Weinberg equilibrium (HWE) using the HWE Institute of Human Genetics calculator (https://ihg.gsf.de/ihg/index_engl.html) for both cases and controls through the Pearson $\chi^2$ test. All SNPs were analyzed using a codominant, dominant, and recessive model to examine the relationships between the two groups (T2DM and non-DM) using the "SNPassoc" package of the R version 3.3.1 (R Foundation, Vienna, Austria). All statistical tests performed in this study were two-tailed, and a significance level of $P<0.05$ was considered statistically significant.

## Results

Among all 5,011 participants, predominantly women (3,854 [77%]) with an average age of 35.44 ± 8.24 years, the study revealed that 51% of participants were overweight or obese (Table 1). Notably, men showed a significantly higher prevalence of obesity, abdominal obesity, and HTN compared to women. Men were also more likely to have an incidence of T2DM (elevated FBG and/or HbA1c) and DLP (high TC, TG, and LDL-C, but low HDL-C) compared to women ($P<0.001$).

A total of 1,464 participants (29.2%) met the criteria for MetS, with a higher prevalence among men (544 [47.0%]) than women (920 [23.9%]). Furthermore, 2,540 participants (50.6%) reported regular exercise (at least once a week), and 3,168 (63.2%) reported monthly alcohol consumption. During follow-up, data from 4,038 participants showed significant increases in body mass index (BMI), WC, SBP, DBP, FBG, MetS, and LDL-C levels, particularly among women. Meanwhile, eGFR and Cr levels declined over time. We primarily collected follow-up data through routine electronic health check-ups, limiting access to key information like HbA1c levels, underlying diseases, family history, and lifestyle factors. Due to the nature of these check-ups, we couldn't gather extra data beyond what's routinely collected in healthcare. While participants were physically present, data collection was restricted by standard procedures during these visits.

Among the participants, 279 (6%) were identified with T2DM. Remarkably, 3,487 participants (72%) who denied having T2DM had normal HbA1c levels, while 1,055 (22%) had unaware pre-DM. Additionally, 70 participants (1.5%) had unaware T2DM (HbA1c > 6.4%) (S1 Fig). The factor-stratified prevalence of diabetes status highlighted higher pre-DM and T2DM rates among older adults, particularly those aged ≥ 40 years, who were associated with being overweight, obese, smoking, and alcohol consumption (Table 2). Most individuals with pre-DM and T2DM had concurrent HTN and DLP, indicated by elevated SBP, TG and LDL-C, but decreased HDL-C. Among those with T2DM, 12.2% had microalbuminuria, and 5.9% had macroalbuminuria. These findings emphasize the importance of health promotion

**Table 1. Demographic data of the participants in the cohort.**

| Characteristics (n (%) or mean±SD) | Baseline (SIH1 and SIH2) | | | | Follow-up (SIH1) | | | | P-value[c] |
|---|---|---|---|---|---|---|---|---|---|
| | Total (N = 5,011) | Men (n = 1,157) | Women (n = 3,854) | P-value[a] | Total (N = 4,034) | Men (n = 925) | Women (n = 3,109) | P-value[b] | |
| Age, years | 35.44 ± 8.24 | 35.60 ± 7.85 | 35.40 ± 8.35 | 0.491 | 41.00 ± 0.13 | 41.11 ± 0.26 | 40.96 ± 0.15 | 0.641 | <0.001 |
| < 30 | 1,415 (28.2) | 295 (25.5) | 1,120 (29.1) | 0.020 | 237 (5.9) | 44 (4.8) | 193 (6.2) | 0.079 | |
| 30–39 | 2,073 (41.4) | 515 (44.5) | 1,558 (40.4) | | 1,646 (40.8) | 374 (40.4) | 1,272 (40.9) | | |
| 40–49 | 1,129 (22.5) | 269 (23.2) | 860 (22.3) | | 1,466 (36.3) | 363 (39.2) | 1,103 (35.5) | | |
| ≥ 50 | 394 (7.9) | 78 (6.7) | 316 (8.2) | | 685 (17.0) | 144 (15.6) | 541 (17.4) | | |
| **Physical examinations** | | | | | | | | | |
| BMI, kg/m² | | | | | | | | | 0.036 |
| < 18.5 | 351 (7.6) | 23 (2.2) | 328 (9.2) | <0.001 | 237 (6.4) | 18 (2.2) | 219 (7.6) | <0.001 | |
| 18.5–22.9 | 1,903 (41.2) | 285 (26.8) | 1,618 (45.5) | | 1,463 (39.6) | 202 (25.0) | 1,261 (43.7) | | |
| 23.0–24.9 | 728 (15.8) | 241 (22.7) | 487 (13.7) | | 627 (17.0) | 178 (22.0) | 449 (15.6) | | |
| 25.0–29.9 | 1,138 (24.6) | 364 (34.2) | 774 (21.8) | | 919 (24.9) | 288 (35.6) | 631 (21.9) | | |
| > 29.9 | 501 (10.8) | 150 (14.1) | 351 (9.9) | | 446 (12.1) | 123 (15.2) | 323 (11.2) | | |
| WC, cm | | | | | | | | | <0.001 |
| men < 90, women < 80 | 3,267 (65.4) | 729 (63.3) | 2,538 (66.0) | 0.489 | 2,346 (63.5) | 505 (62.4) | 1,841 (63.9) | 0.454 | |
| men ≥ 90, women ≥ 80 | 1,728 (34.6) | 423 (36.7) | 1,305 (34.0) | | 1,346 (36.5) | 304 (37.6) | 1,042 (36.1) | | |
| SBP, mmHg | | | | | | | | | <0.001 |
| < 120 | 4,015 (80.3) | 719 (62.3) | 3,296 (85.7) | <0.001 | 2,898 (78.5) | 480 (59.3) | 2,418 (83.9) | <0.001 | |
| 120–139 | 619 (12.4) | 270 (23.4) | 349 (9.1) | | 574 (15.5) | 228 (28.2) | 346 (12.0) | | |
| 140–159 | 316 (6.3) | 147 (12.7) | 169 (4.4) | | 199 (5.4) | 95 (11.7) | 104 (3.6) | | |
| ≥ 160 | 48 (1.0) | 18 (1.6) | 30 (0.8) | | 21 (0.6) | 6 (0.7) | 15 (0.5) | | |
| DBP, mmHg | | | | | | | | | <0.001 |
| < 80 | 3,918 (78.4) | 791 (68.5) | 3,127 (81.3) | <0.001 | 2,919 (79.1) | 539 (66.6) | 2,380 (82.6) | <0.001 | |
| 80–89 | 776 (15.5) | 245 (21.2) | 531 (13.8) | | 616 (16.7) | 210 (26.0) | 406 (14.1) | | |
| 90–99 | 236 (4.7) | 89 (7.7) | 147 (3.8) | | 134 (3.6) | 53 (6.6) | 81 (2.8) | | |
| ≥ 100 | 68 (1.4) | 29 (2.5) | 39 (1.0) | | 23 (0.6) | 7 (0.9) | 16 (0.6) | | |
| **Laboratory measurements** | | | | | | | | | |
| FBG, mg/dL | | | | | | | | | <0.001 |
| < 100 | 4,486 (89.6) | 959 (83.0) | 3,527 (91.6) | | 2,979 (86.5) | 587 (80.6) | 2,392 (88.1) | <0.001 | |
| 100–125 | 431 (8.6) | 169 (14.6) | 262 (6.8) | | 368 (10.7) | 115 (15.8) | 253 (9.3) | | |
| > 125 | 87 (1.7) | 27 (2.3) | 60 (1.6) | | 97 (2.8) | 26 (3.6) | 71 (2.6) | | |
| HbA1c, % | | | | | | | | | n/a |
| < 5.7 | 3,688 (73.7) | 718 (62.1) | 2,970 (77.2) | <0.001 | n/a | n/a | n/a | n/a | |
| 5.7–6.4 | 1,157 (23.1) | 388 (33.5) | 769 (20.0) | | n/a | n/a | n/a | | |
| > 6.4 | 161 (3.2) | 51 (4.4) | 110 (2.9) | | n/a | n/a | n/a | | |
| TC, mg/dL | | | | | | | | | 0.053 |
| < 200 | 3,158 (63.1) | 650 (56.3) | 2,508 (65.2) | <0.001 | 2,106 (61.4) | 403 (55.5) | 1,703 (63.0) | 0.001 | |
| ≥ 200 | 1,846 (36.9) | 505 (43.7) | 1,341 (34.8) | | 1,322 (38.6) | 323 (44.5) | 999 (37.0) | | |
| TG, mg/dL | | | | | | | | | 0.058 |
| < 150 | 4,316 (86.3) | 865 (74.9) | 3,451 (89.7) | <0.001 | 2,903 (84.6) | 525 (72.2) | 2,378 (87.9) | <0.001 | |
| ≥ 150 | 688 (13.7) | 290 (25.1) | 398 (10.3) | | 528 (15.4) | 202 (27.8) | 326 (12.1) | | |
| HDL-C, mg/dL | | | | | | | | | 0.044 |
| men < 40, women < 50 | 1,251 (25.0) | 554 (48.0) | 697 (18.1) | <0.001 | 835 (24.3) | 346 (47.5) | 489 (18.1) | <0.001 | |
| men ≥ 40, women ≥ 50 | 3,753 (75.0) | 601 (52.0) | 3,152 (81.9) | | 2,595 (75.7) | 382 (52.5) | 2,213 (81.9) | | |
| LDL-C, mg/dL | | | | | | | | | <0.001 |
| < 130 | 3,833 (76.6) | 744 (64.5) | 3,089 (80.3) | <0.001 | 2,597 (75.9) | 476 (65.7) | 2,121 (78.6) | <0.001 | |
| 130–159 | 874 (17.5) | 294 (25.5) | 580 (15.1) | | 604 (17.7) | 178 (24.6) | 426 (15.8) | | |
| > 159 | 295 (5.9) | 115 (10.0) | 180 (4.7) | | 220 (6.4) | 70 (9.7) | 150 (5.6) | | |
| MetS | | | | | | | | | <0.001 |
| no | 3,547 (70.8) | 613 (53.0) | 2934 (76.1) | <0.001 | 2,339 (68.2) | 419 (52.5) | 1,920 (72.9) | <0.001 | |
| yes | 1,464 (29.2) | 544 (47.0) | 920 (23.9) | | 1,091 (31.8) | 379 (47.5) | 712 (27.1) | | |

(*Continued*)

**Table 1.** (Continued)

| Characteristics (n (%) or mean±SD) | Baseline (SIH1 and SIH2) | | | | Follow-up (SIH1) | | | | P-value[c] |
|---|---|---|---|---|---|---|---|---|---|
| | Total (N = 5,011) | Men (n = 1,157) | Women (n = 3,854) | P-value[a] | Total (N = 4,034) | Men (n = 925) | Women (n = 3,109) | P-value[b] | |
| eGFR, mL/min/1.73m² | 102.34 ± 14.65 | 95.33 ± 14.84 | 104.64 ± 13.84 | <0.001 | 100.57 ± 14.78 | 93.43 ± 15.08 | 102.67 ± 14.02 | <0.001 | <0.001 |
| Cr, mg/dL | 24.54 ± 66.10 | 25.84 ± 70.31 | 24.12 ± 64.67 | 0.451 | 18.28 ± 55.23 | 20.27 ± 61.61 | 17.53 ± 52.66 | 0.513 | <0.001 |
| MAU/Cr ratio, mg/g | 14.80 ± 96.32 | 13.65 ± 121.25 | 15.17 ± 86.91 | 0.734 | n/a | n/a | n/a | n/a | n/a |
| Hemoglobin, g/dL | 13.15 ± 1.47 | 14.81 ± 1.15 | 12.66 ± 1.15 | <0.001 | 13.02 ± 1.43 | 14.70 ± 1.10 | 12.57 ± 1.15 | <0.001 | 0.054 |
| Hematocrit, % | 40.37 ± 4.00 | 45.05 ± 3.00 | 38.99 ± 3.13 | <0.001 | 39.75 ± 3.88 | 44.37 ± 2.92 | 38.53 ± 3.09 | <0.001 | 0.408 |
| **Self-report questionnaires** | | | | | | | | | |
| **Underlying diseases** | | | | | | | | | |
| DM, yes | 209 (4.6) | 54 (4.6) | 155 (4.0) | 0.278 | n/a | n/a | n/a | n/a | n/a |
| HTN, yes | 287 (6.4) | 96 (8.3) | 191 (4.9) | <0.001 | n/a | n/a | n/a | n/a | n/a |
| DLP, yes | 561 (12.4) | 180 (15.5) | 381 (9.8) | <0.001 | n/a | n/a | n/a | n/a | n/a |
| Ischemic heart, yes | 21 (0.5) | 5 (0.4) | 16 (0.4) | 0.173 | n/a | n/a | n/a | n/a | n/a |
| Stroke, yes | 11 (0.2) | 3 (0.3) | 8 (0.2) | 0.005 | n/a | n/a | n/a | n/a | n/a |
| Gout, yes | 69 (1.5) | 48 (4.2) | 21 (0.5) | <0.001 | n/a | n/a | n/a | n/a | n/a |
| **Family history** | | | | | | | | | |
| Heart failure, yes | 293 (6.5) | 72 (6.2) | 221 (5.7) | 0.439 | n/a | n/a | n/a | n/a | n/a |
| Cancer, yes | 991 (21.9) | 187 (16.2) | 804 (20.8) | <0.001 | n/a | n/a | n/a | n/a | n/a |
| Stroke, yes | 333 (7.3) | 67 (5.7) | 266 (6.9) | 0.119 | n/a | n/a | n/a | n/a | n/a |
| **Lifestyle** | | | | | | | | | |
| Exercise, yes | 2,540 (50.6) | 740 (63.9) | 1,800 (46.7) | 0.001 | n/a | n/a | n/a | n/a | n/a |
| Smoking, yes | 306 (6.1) | 225 (19.4) | 81 (2.1) | 0.001 | n/a | n/a | n/a | n/a | n/a |
| Alcohol consumption, yes | 3,168 (63.2) | 953 (82.4) | 2,215 (57.4) | 0.013 | n/a | n/a | n/a | n/a | n/a |

Continuous variables were presented as mean±standard deviation. Categorical variables were presented as number (percentages).

[a,b]P-value is the differences between men and women based on independent t-test and chi-square for continuous and categorical variables, respectively.

[c]P-value is the differences between baseline and follow-up based on Wilcoxon signed rank test and McNemar chi-square test for continuous and categorical variables, respectively.

n/a, not available; BMI, body mass index; WC, waist circumference; SBP, systolic blood pressure; DBP, diastolic blood pressure; FBG, fasting blood glucose; HbA1c, glycated hemoglobin; TC, total cholesterol; TG, triglycerides; HDL-C, high-density lipoprotein cholesterol; LDL-C, low-density lipoprotein cholesterol; MetS, metabolic syndrome; eGFR, estimated glomerular filtration rate; Cr, creatinine; MAU/Cr, urine albumin-to-creatinine ratio; DM, diabetes mellitus; HTN, hypertension; DLP, dyslipidemia

efforts to prevent T2DM and related complications among the working-age population since these are modifiable.

A cross-sectional analysis comparing pre-DM and T2DM to those without T2DM at baseline was conducted using a multivariate logistic regression model (Table 3). Sex, age, BMI, HTN, DLP, gout, alcohol consumption, exercise, and smoking were adjusted. Women displayed an almost two-fold increased odds (OR = 1.71, 95% CI: 1.37–2.13) of having pre-DM compared to men. Pre-DM was significantly associated with sex, older age, overweight/obesity, elevated WC, having MetS, elevated LDL-C concentrations, gout, and being non-exercise. Participants aged 40 years or older displayed three-fold increased odds of having pre-DM (OR = 3.60, 95% CI: 2.86–4.55) and T2DM (OR = 3.28, 95% CI: 2.09–5.14). Participants with pre-DM displayed the metabolic profile defined by overweight (OR = 1.45, 95% CI: 1.13–1.85), obesity (OR = 2.15, 95% CI: 1.66–2.78), having MetS (OR = 1.50, 95% CI: 1.10–1.98), and elevated LDL-C concentrations (OR = 1.30, 95% CI: 1.01–1.67). Normal HDL-C concentrations

**Table 2. Characteristics of the participants based on self-report diabetes status.**

| Characteristics | | All | | Diabetes status | | | | | | P-value [d] | | |
|---|---|---|---|---|---|---|---|---|---|---|---|---|
| | | | | Non-DM [a] | | Unaware Pre-DM [b] | | T2DM [c] | | | | |
| | | (N = 4,821) | | (n = 3,487) | | (n = 1,055) | | (n = 279) | | | | |
| | | % | (95% CI) | % | (95% CI) | % | (95% CI) | % | (95% CI) | A | B | C |
| Sex | | | | | | | | | | <0.001 | <0.001 | 0.159 |
| | men | 22.2 | (20.1–24.5) | 18.6 | (16.4–27.2) | 32.4 | (25.9–64.0) | 27.9 | (15.1–42.5) | | | |
| | women | 77.8 | (59.9–85.0) | 81.4 | (62.7–99.8) | 67.6 | (52.1–91.2) | 72.1 | (55.5–97.2) | | | |
| Age, years | | | | | | | | | | <0.001 | <0.001 | 0.015 |
| | < 30 | 27.9 | (26.6–29.3) | 33.2 | (31.6–34.9) | 14.6 | (12.5–17.0) | 12.1 | (8.5–16.7) | | | |
| | 30–39 | 41.9 | (40.4–43.4) | 43.8 | (42.1–45.5) | 38.6 | (35.6–41.7) | 30.9 | (25.4–36.9) | | | |
| | ≥ 40 | 30.2 | (28.8–31.6) | 22.9 | (21.5–24.4) | 46.7 | (43.5–49.8) | 57.0 | (50.8–63.0) | | | |
| BMI, kg/m$^2$ | | | | | | | | | | <0.001 | <0.001 | <0.001 |
| | normal (< 23.0) | 49.6 | (48.1–51.1) | 58.9 | (57.2–60.7) | 27.0 | (26.6–29.3) | 19.8 | (15.3–25.1) | | | |
| | overweight (23.0–24.9) | 15.9 | (14.8–16.9) | 15.9 | (15.3–25.1) | 16.4 | (14.6–17.2) | 14.1 | (10.2–19.0) | | | |
| | obesity (≥ 25.0) | 34.5 | (33.1–36.8) | 25.2 | (23.7–26.8) | 56.7 | (53.5–59.8) | 66.1 | (60.0–71.7) | | | |
| WC, cm | | | | | | | | | | <0.001 | <0.001 | <0.001 |
| | normal (men < 90, women < 80) | 66.1 | (64.6–67.5) | 75.1 | (73.6–76.6) | 45.8 | (42.6–48.9) | 30.6 | (25.2–36.6) | | | |
| | elevated (men ≥ 90, women ≥ 80) | 33.9 | (32.4–35.3) | 24.9 | (23.4–26.4) | 54.2 | (51.0–57.3) | 69.4 | (63.3–74.7) | | | |
| BP, mmHg | | | | | | | | | | <0.001 | <0.001 | 0.014 |
| | normal (< 120/80) | 61.7 | (60.2–63.1) | 68.2 | (66.5–69.8) | 45.7 | (42.6–48.9) | 41.9 | (35.9–48.1) | | | |
| | elevated (≥ 120/80) | 38.3 | (36.8–39.7) | 31.8 | (30.1–33.4) | 54.2 | (51.0–57.3) | 58.1 | (51.8–64.0) | | | |
| SBP, mmHg | | | | | | | | | | <0.001 | <0.001 | 0.335 |
| | normal (< 120) | 60.6 | (59.6–66.2) | 66.9 | (62.8–70.3) | 44.6 | (41.2–49.9) | 41.4 | (39.6–52.5) | | | |
| | elevated (≥ 120) | 39.4 | (30.2–45.1) | 33.1 | (30.6–38.2) | 55.4 | (51.8–58.4) | 58.6 | (49.6–64.3) | | | |
| DBP, mmHg | | | | | | | | | | <0.001 | <0.001 | 0.049 |
| | normal (< 80) | 78.7 | (69.4–82.9) | 83.3 | (79.5–85.6) | 68.1 | (66.7–71.3) | 61.9 | (59.6–62.1) | | | |
| | elevated (≥ 80) | 21.3 | (18.9–22.4) | 16.7 | (14.8–17.3) | 31.9 | (29.5–33.1) | 38.1 | (35.4–42.5) | | | |
| TC, mg/dL | | | | | | | | | | <0.001 | 0.018 | 0.074 |
| | normal (< 200) | 63.8 | (62.4–65.2) | 67.4 | (65.6–68.9) | 53.3 | (50.0–56.4) | 59.8 | (55.2–61.4) | | | |
| | elevated (≥ 200) | 36.2 | (34.7–37.6) | 32.6 | (31.0–34.3) | 46.7 | (43.5–49.9) | 40.2 | (39.5–44.9) | | | |
| TG, mg/dL | | | | | | | | | | <0.001 | <0.001 | <0.001 |
| | normal (< 150) | 86.6 | (85.5–87.5) | 91.1 | (90.0–92.0) | 77.3 | (74.5–79.8) | 65.5 | (64.5–69.8) | | | |
| | elevated (≥ 150) | 13.4 | (12.4–14.4) | 8.9 | (7.9–9.9) | 22.7 | (20.1–25.4) | 34.5 | (30.1–36.4) | | | |
| HDL-C, mg/dL | | | | | | | | | | <0.001 | <0.001 | 0.344 |
| | normal (men ≥ 40, women ≥ 50) | 24.4 | (23.1–25.6) | 18.5 | (17.1–19.9) | 38.8 | (35.7–41.9) | 42.2 | (35.7–44.9) | | | |
| | low (men < 40, women < 50) | 75.7 | (74.3–76.9) | 81.5 | (80.0–82.8) | 61.2 | (58.1–64.2) | 57.8 | (56.1–64.2) | | | |
| LDL-C, mg/dL | | | | | | | | | | <0.001 | <0.001 | <0.001 |
| | normal (< 130) | 77.1 | (75.8–78.3) | 80.9 | (79.5–82.2) | 65.2 | (62.1–68.1) | 75.0 | (62.1–78.1) | | | |
| | elevated (≥ 130) | 22.9 | (21.6–24.1) | 19.1 | (17.7–20.4) | 34.8 | (31.8–37.8) | 25.0 | (21.8–37.8) | | | |
| MAU/Cr ratio, mg/g | | | | | | | | | | 0.013 | <0.001 | <0.001 |
| | normal (< 30) | 95.2 | (94.4–95.8) | 96.6 | (95.8–97.1) | 94.2 | (92.4–95.5) | 81.9 | (80.4–95.5) | | | |
| | microalbuminuria (30–299) | 3.9 | (3.3–4.6) | 2.8 | (2.2–3.5) | 5.4 | (4.0–7.0) | 12.2 | (10.0–17.0) | | | |
| | macroalbuminuria (> 299) | 0.9 | (0.6–1.2) | 0.6 | (0.3–0.9) | 0.5 | (0.1–1.1) | 5.9 | (1.1–6.1) | | | |

(*Continued*)

**Table 2.** (Continued)

| Characteristics | | All | | Diabetes status | | | | | | P-value [d] | | |
| --- | --- | --- | --- | --- | --- | --- | --- | --- | --- | --- | --- | --- |
| | | | | Non-DM[a] | | Unaware Pre-DM[b] | | T2DM[c] | | | | |
| | | (N = 4,821) | | (n = 3,487) | | (n = 1,055) | | (n = 279) | | | | |
| | | % | (95% CI) | % | (95% CI) | % | (95% CI) | % | (95% CI) | A | B | C |
| MetS | | | | | | | | | | <0.001 | <0.001 | <0.001 |
| | no | 76.7 | (74.3–79.0) | 53.4 | (50.4–55.9) | 61.3 | (59.7–63.7) | 85.3 | (80.4–89.2) | | | |
| | yes | 28.3 | (26.9–29.6) | 46.6 | (43.7–49.4) | 38.7 | (35.9–41.4) | 14.7 | (12.7–16.8) | | | |
| **Underlying diseases** | | | | | | | | | | | | |
| HTN, yes | | 5.2 | (4.1–7.1) | 2.5 | (1.9–3.4) | 7.2 | (5.5–9.6) | 31.3 | (24.1–42.2) | <0.001 | <0.001 | <0.001 |
| DLP, yes | | 10.8 | (8.3–14.5) | 7.0 | (5.4–9.4) | 14.8 | (11.3–19.9) | 42.6 | (32.9–57.5) | <0.001 | <0.001 | <0.001 |
| **Lifestyle** | | | | | | | | | | | | |
| Exercise, yes | | 50.8 | (39.1–68.5) | 50.1 | (38.6–67.6) | 54.6 | (42.0–73.6) | 44.5 | (34.3–60.0) | 0.038 | 0.335 | 0.033 |
| Smoking, yes | | 5.5 | (4.7–9.8) | 3.1 | (2.3–4.1) | 4.9 | (3.8–6.6) | 4.9 | (3.7–6.6) | 0.002 | 0.001 | 0.099 |
| Alcohol consumption, yes | | 62.9 | (50.5–78.9) | 61.4 | (51.9–65.6) | 65.1 | (54.9–68.8) | 62.8 | (59.8–74.3) | 0.572 | 0.023 | 0.035 |

[a]Non-DM was defined as a self-reported medical history of no diabetes mellitus and HbA1c level < 5.7%

[b]Unaware Pre-DM was defined as a self-reported medical history of no diabetes mellitus and HbA1c level 5.7–6.4%

[c]T2DM were defined as a self-reported medical history of no diabetes mellitus and HbA1c > 6.4%, and/or self-reported medical history of diabetes mellitus.

[d]P-value compares diabetes status by using the chi-square test. (A) P-value compared non-DM and unaware pre-DM individuals, (B) P-value compared non-DM and T2DM individuals

(C) P-value compared unaware pre-DM and T2DM individuals. Data are presented as 95 confidence interval (95%CI) and percentage.

BMI, body mass index; WC, waist circumference; BP, blood pressure; SBP, systolic blood pressure; DBP, diastolic blood pressure; TC, total cholesterol; TG, triglycerides; HDL-C, high density lipoprotein cholesterol; LDL-C, low density lipoprotein cholesterol; MetS, metabolic syndrome; MAU/Cr, urine albumin-to-creatinine ratio; DM, diabetes mellitus; HTN, hypertension; DLP, dyslipidemia

was protective against having pre-DM (OR = 0.78, 95% CI: 0.61–0.99). Associations with older age, having MetS, HTN, and DLP were noticeably significant ($P > 0.001$) among participants diagnosed with T2DM. Obese participants displayed a 75% increased odds (OR = 1.75, 95% CI: 1.03–3.00) of having T2DM. Participants meeting the criteria for MetS according to the IDF definition displayed five-fold increased odds (OR = 5.11, 95% CI: 3.09–8.44) of having T2DM. Participants with elevated TG displayed a 64% increased odds (OR = 1.64, 95% CI: 1.11–2.42) of having T2DM. With regards to lifestyle behaviors, being a smoker was significantly associated with T2DM (OR = 1.69, 95% CI: 1.03–2.77) while lack of exercise was associated with pre-DM (OR = 1.21, 95% CI: 1.03–1.43). The risk factors showing significant association among participants diagnosed with pre-DM or T2DM are displayed in Fig 2.

The longitudinal analysis of the association between the changes in risk factors from baseline to follow-up (SIH1 only) and the relative risk of developing pre-DM or T2DM were analyzed using a multivariate multinomial logistic regression model (Table 4). Notably, several factors were significantly associated with the development of pre-DM. These included being aged 45 years or older (RRR = 1.96, 95% CI: 1.10–3.47), experiencing an increase in WC (RRR = 2.54, 95% CI: 1.12–5.75), developing MetS (RRR = 4.11, 95% CI: 2.23–5.58), remaining MetS (RRR = 2.27, 95% CI: 1.20–4.30), experiencing an increase in TG levels (RRR = 3.95, 95% CI: 1.86–8.36), remaining elevated TG levels (RRR = 5.01, 95% CI: 1.85–9.54). Interestingly, participants who maintained elevated TG exhibited even stronger and statistically significant associations with the development of T2DM (RRR = 5.27, 95% CI: 2.22–8.51). Furthermore, participants who experienced an increase in WC were significantly associated with the development of pre-DM but not T2DM. It is noteworthy that the certain risk factors associated with both the development of pre-DM and T2DM were older age, the development

**Table 3. Cross-sectional analysis comparing pre-diabetes[a] (n = 1,055) and type 2 diabetes mellitus[b] (n = 279) to those without diabetes[c] (n = 3,487) at baseline (SIH1 and SIH2).**

| Factors | | Risk factors for Pre-DM[a] | | Risk factors for T2DM[b] | |
|---|---|---|---|---|---|
| | | OR (95% CI) | P-value[d] | OR (95% CI) | P-value[d] |
| Sex | | | | | |
| | men | 1.00 | | 1.00 | |
| | women | 1.71 (1.37–2.13) | <0.001 | 1.01 (0.66–1.54) | 0.954 |
| Age, years | | | | | |
| | < 30 | 1.00 | | 1.00 | |
| | 30–39 | 1.68 (1.34–2.10) | <0.001 | 1.30 (0.82–2.06) | 0.260 |
| | ≥ 40 | 3.60 (2.86–4.55) | <0.001 | 3.28 (2.09–5.14) | <0.001 |
| BMI, kg/m$^2$ | | | | | |
| | normal (< 23.0) | 1.00 | | 1.00 | |
| | overweight (23.0–24.9) | 1.45 (1.13–1.85) | 0.003 | 1.23 (0.73–2.08) | 0.442 |
| | obese (≥ 25.0) | 2.15 (1.66–2.78) | <0.001 | 1.75 (1.03–3.00) | 0.040 |
| WC, cm | | | | | |
| | normal (men < 90, women < 80) | 1.00 | | 1.00 | |
| | elevated (men ≥ 90, women ≥ 80) | 1.38 (1.07–1.78) | 0.012 | 1.34 (0.82–2.21) | 0.245 |
| BP, mmHg | | | | | |
| | normal (< 120/80) | 1.00 | | 1.00 | |
| | elevated (≥ 120/80) | 1.18 (0.98–1.42) | 0.084 | 0.79 (0.55–1.12) | 0.183 |
| MetS | | | | | |
| | no | 1.00 | | 1.00 | |
| | yes | 1.50 (1.1–1.98) | 0.005 | 5.11 (3.09–8.44) | <0.001 |
| TC, mg/dL | | | | | |
| | normal (< 200) | 1.00 | | 1.00 | |
| | elevated (≥ 200) | 1.19 (0.95–1.50) | 0.135 | 1.34 (0.89–2.00) | 0.163 |
| TG, mg/dL | | | | | |
| | normal (< 150) | 1.00 | | 1.00 | |
| | elevated (≥ 150) | 1.27 (0.99–1.64) | 0.059 | 1.64 (1.11–2.42) | 0.013 |
| HDL-C, mg/dL | | | | | |
| | low (men < 40, women < 50) | 1.00 | | 1.00 | |
| | normal (men ≥ 40, women ≥ 50) | 0.78 (0.61–0.99) | 0.041 | 1.32 (0.90–1.94) | 0.149 |
| LDL-C, mg/dL | | | | | |
| | normal (< 130) | 1.00 | | 1.00 | |
| | elevated (≥ 130) | 1.30 (1.01–1.67) | 0.039 | 0.69 (0.44–1.10) | 0.117 |
| HTN | | | | | |
| | no | 1.00 | | 1.00 | |
| | yes | 1.03 (0.81–1.31) | 0.808 | 2.34 (1.75–3.14) | <0.001 |
| DLP | | | | | |
| | no | 1.00 | | 1.00 | |
| | yes | 0.95 (0.81–1.11) | 0.523 | 1.72 (1.36–2.18) | <0.001 |
| Gout | | | | | |
| | no | 1.00 | | 1.00 | |
| | yes | 1.27 (1.01–1.59) | 0.038 | 0.91 (0.63 to 1.31) | 0.607 |
| Exercise | | | | | |
| | no | 1.00 | | 1.00 | |
| | yes | 1.21 (1.03–1.43) | 0.022 | 0.93 (0.69 to 1.26) | 0.655 |
| Smoking | | | | | |

(*Continued*)

**Table 3.** (Continued）

| Factors | | Risk factors for Pre-DM[a] | | Risk factors for T2DM[b] | |
|---|---|---|---|---|---|
| | | OR (95% CI) | P-value[d] | OR (95% CI) | P-value[d] |
| | no | 1.00 | | 1.00 | |
| | yes | 0.73 (0.54–0.97) | 0.132 | 1.69 (1.03–2.77) | 0.039 |
| Alcohol consumption | | | | | |
| | no | 1.00 | | 1.00 | |
| | yes | 1.08 (0.91–1.28) | 0.362 | 0.76 (0.55–1.05) | 0.098 |

[a]Pre-DM was defined as a HbA1c level between 5.7 and 6.4%.

[b]T2DM was defined as a self-reported medical history of diabetes, current use of hypoglycemic medication, and/or a HbA1c level > 6.4%.

[c]Non-DM was defined as a self-reported medical history of no-DM and HbA1c level < 5.7%.

[d]P-value < 0.05 for test of null hypothesis that the odds ratio is equal to the odds ratio in the reference category.

Sex, age, BMI, HTN, DLP, gout, alcohol consumption, exercise, and smoking were adjusted.

OR was odd ratio with 95% confidence intervals (CI).

SIH, Siriraj Health; pre-DM, pre-diabetes; T2DM, type 2 diabetes mellitus; BMI, body mass index; WC, waist circumference; BP, blood pressure; MetS, metabolic syndrome; TC, total cholesterol; TG, triglycerides; HDL-C, high-density lipoprotein cholesterol; LDL-C, low-density lipoprotein cholesterol; HTN, hypertension; DLP, dyslipidemia

of MetS, the maintenance of MetS, an increase in TG, and the maintenance of elevated TG. Participants aged 45 years or older displayed an 83% increased relative risks of developing T2DM (RRR = 1.83, 95% CI: 1.10–3.05). Participants who developed MetS displayed a three-fold increased relative risk of developing T2DM (RRR = 3.10, 95% CI: 1.77–5.43), while participants with maintained MetS displayed a two-fold increased relative risk of developing T2DM (RRR = 2.00, 95% CI: 1.13–3.54). Associations with increases in TG concentrations, whether

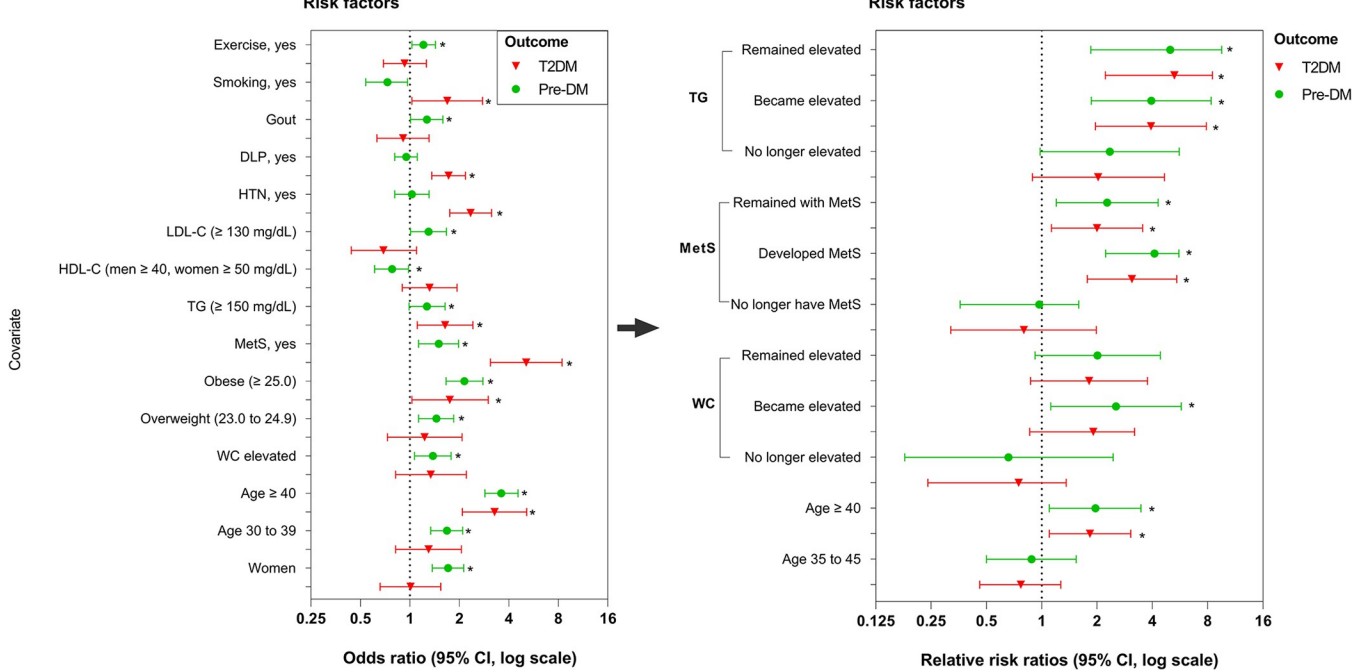

**Fig 2. Comparative analysis of pre-diabetes (Pre-DM) and type 2 diabetes (T2DM) to those without T2DM at baseline, and risk assessment for Pre-DM and T2DM development during follow-up, DLP, dyslipidemia; HTN, hypertension; LDL-C, low-density lipoprotein cholesterol; HDL-C, high-density lipoprotein cholesterol; TG, triglycerides; MetS, metabolic syndrome; WC, waist circumference.**

**Table 4. Changes in risk factors from baseline (SIH1) and follow-up (SIH1) and relative risk of developing pre-diabetes[a] (n = 120) or type 2 diabetes mellitus[b] (n = 48) in follow-up among individuals with normal blood glucose levels[c] (N = 2,309) at baseline.**

| Factors | | Overall (n = 2,477) | Relative risk ratios (RRRs) for individuals who developed pre-DM during the follow-up period[a] | | Relative risk ratios (RRRs) for individuals who developed T2DM during the follow-up period[b] | |
|---|---|---|---|---|---|---|
| | | | RRR (95% CI) | P-value[d] | RRR (95% CI) | P-value[d] |
| Sex | | | | | | |
| | men | 451 | 1.00 | | 1.00 | |
| | women | 2,026 | 0.92 (0.51–1.64) | 0.769 | 1.03 (0.61–1.74) | 0.906 |
| Age, years | | | | | | |
| | < 35 | 848 | 1.00 | | 1.00 | |
| | 35–45 | 1,059 | 0.88 (0.50–1.54) | 0.659 | 0.77 (0.46–1.27) | 0.298 |
| | $\geq$ 45 | 570 | 1.96 (1.10–3.47) | 0.022 | 1.83 (1.10–3.05) | 0.020 |
| BMI, kg/m$^2$ | | | | | | |
| | Persistently normal | 1,628 | 1.00 | | 1.00 | |
| | Becoming normal | 91 | 1.75 (0.62–4.95) | 0.292 | 1.87 (0.72–4.88) | 0.200 |
| | Persistently overweight | 361 | 0.90 (0.42–1.92) | 0.788 | 0.96 (0.47–1.97) | 0.909 |
| | Persistently obese | 128 | 0.73 (0.25–2.16) | 0.573 | 1.05 (0.41–2.66) | 0.924 |
| | Becoming overweight/obese | 160 | 0.83 (0.34–2.03) | 0.681 | 0.82 (0.35–1.91) | 0.647 |
| WC, cm | | | | | | |
| | Never elevated | 1,587 | 1.00 | | 1.00 | |
| | No longer elevated | 109 | 0.66 (0.18–2.45) | 0.538 | 0.75 (0.24–1.36) | 0.623 |
| | Became elevated | 146 | 2.54 (1.12–5.75) | 0.025 | 1.91 (0.86–4.20) | 0.110 |
| | Remained elevated | 530 | 2.01 (0.92–4.42) | 0.081 | 1.81 (0.87–3.76) | 0.110 |
| BP, mmHg | | | | | | |
| | Never elevated | 2,187 | 1.00 | | 1.00 | |
| | No longer elevated | 74 | 0.92 (0.20–4.18) | 0.917 | 1.62 (0.42–6.30) | 0.486 |
| | Became elevated | 108 | 0.61 (0.23–1.62) | 0.325 | 0.80 (0.32–2.00) | 0.636 |
| | Remained elevated | 108 | 0.66 (0.19–2.29) | 0.511 | 1.67 (0.56–4.98) | 0.361 |
| MetS | | | | | | |
| | Never had MetS | 1,206 | 1.00 | | 1.00 | |
| | No longer have MetS | 210 | 0.97 (0.36–1.59) | 0.952 | 0.80 (0.32–1.98) | 0.629 |
| | Developed MetS | 378 | 4.11 (2.23–5.58) | <0.001 | 3.10 (1.77–5.43) | <0.001 |
| | Remained with MetS | 577 | 2.27 (1.20–4.30) | 0.012 | 2.00 (1.13–3.54) | 0.017 |
| TC, mg/dL | | | | | | |
| | Never elevated | 1,331 | 1.00 | | 1.00 | |
| | No longer elevated | 221 | 0.96 (0.41–2.28) | 0.931 | 0.81 (0.37–1.78) | 0.601 |
| | Became elevated | 297 | 1.72 (0.84–3.53) | 0.138 | 1.46 (0.76–2.78) | 0.252 |
| | Remained elevated | 593 | 1.29 (0.60–2.76) | 0.508 | 0.96 (0.48–1.93) | 0.917 |
| TG, mg/dL | | | | | | |
| | Never elevated | 2,045 | 1.00 | | 1.00 | |
| | No longer elevated | 125 | 2.35 (0.98–5.61) | 0.054 | 2.03 (0.89–4.66) | 0.094 |
| | Became elevated | 165 | 3.95 (1.86–8.36) | <0.001 | 3.93 (1.96–7.87) | <0.001 |
| | Remained elevated | 108 | 5.01 (1.85–9.54) | 0.001 | 5.27 (2.22–8.51) | <0.001 |
| HDL-C, mg/dL | | | | | | |
| | Remained elevated | 1,993 | 0.50 (0.23–1.08) | 0.076 | 0.92 (0.45–1.87) | 0.812 |
| | Became elevated | 116 | 0.48 (0.15–1.56) | 0.221 | 0.45 (0.14–1.45) | 0.179 |
| | No longer elevated | 127 | 1.89 (0.76–4.68) | 0.171 | 2.12 (0.86–5.18) | 0.101 |
| | Never elevated | 207 | 1.00 | | 1.00 | |
| Calculated LDL-C, mg/dL | | | | | | |

*(Continued)*

**Table 4.** (Continued)

| Factors | | Overall (n = 2,477) | Relative risk ratios (RRRs) for individuals who developed pre-DM during the follow-up period[a] | | Relative risk ratios (RRRs) for individuals who developed T2DM during the follow-up period[b] | |
|---|---|---|---|---|---|---|
| | | | RRR (95% CI) | P-value[d] | RRR (95% CI) | P-value[d] |
| | Never elevated | 1,751 | 1.00 | | 1.00 | |
| | No longer elevated | 142 | 2.10 (0.91–4.86) | 0.082 | 2.00 (0.91–4.41) | 0.085 |
| | Became elevated | 189 | 1.19 (0.53–2.67) | 0.668 | 1.21 (0.57–2.57) | 0.625 |
| | Remained elevated | 171 | 1.32 (0.54–3.25) | 0.541 | 1.46 (0.63–3.41) | 0.376 |

[a]Pre-DM was defined as a fasting plasma glucose level between 100 and 125 mg/dL

[b]T2DM was defined as a fasting plasma glucose level $\geq$ 126 mg/ dL, and/or was diagnosed with diabetes.

[c]Normal blood glucose levels were defined as fasting plasma glucose concentrations <100 mg/dL.

[d]P-value < 0.05 for test of null hypothesis that the relative risk ratios are equal to the relative risk ratios in the reference category. Sex, age, WC, TC, BMI, and BP were adjusted.

RRR was relative risk ratio with 95% confidence intervals (CI).

SIH, Siriraj Health; pre-DM, pre-diabetes; T2DM, type 2 diabetes mellitus; BMI, body mass index; WC, waist circumference; BP, blood pressure; MetS, metabolic syndrome; TC, total cholesterol; TG, triglycerides; HDL-C, high-density lipoprotein cholesterol; LDL-C, low-density lipoprotein cholesterol

occurring or being maintained, were noticeably significant and strong among participants diagnosed with developing T2DM. The relative risk factors showing significant association among participants diagnosed with developing pre-DM or T2DM are also displayed in Fig 2.

The average predicted probabilities of maintaining normal blood sugar levels, developing pre-DM, or T2DM during follow-up among participants without T2DM at baseline were analyzed by using the Stata margins command (Fig 3). The lowest probabilities of maintaining normal blood glucose levels and the highest probabilities of progressing to pre-DM or T2DM were observed among individuals who either maintained MetS (0.29, 95% CI: 0.18–0.35), developed MetS (0.19, 95% CI: 0.12–0.28), maintained increases in TG concentrations (0.30, 95% CI: 0.26–0.45), or occurred increases in TG concentrations (0.21, 95% CI: 0.15–0.32).

The distribution of genotypes and allelic frequencies for these TCF7L2 gene polymorphisms was presented in Table 5 and S2 Table in S2 File. Notably, the minor T allele of both rs7903146 (C/T) and rs4506565 (A/T) SNPs showed a trend toward increased risk for developing pre-DM, with odds ratios of 2.74 (95% CI: 0.32–23.3) and 2.71 (95% CI: 0.32–23.07), respectively. However, these associations did not reach statistical significance (P>0.05). Five SNPs (rs12243326, rs12255372, rs4132670, rs4506565, and rs7903146) did not exhibit significant associations with pre-DM and T2DM (S3 Table in S2 File). Additionally, after adjusting for age, it was found that men with the TT genotype of rs7917983 exhibited a 2.81-fold increased risk for T2DM compared to women with the same genotype. Moreover, women with the TT genotype of rs290487 showed a 2.27-fold increased risk for diabetes compared to men with the TT genotype as shown in S3 Table in S2 File.

## Discussion

The SIH study distinguishes itself as a comprehensive, population-based cohort encompassing a diverse spectrum of hospital personnel, comprising healthcare practitioners, supporting staff, and academic faculty. An intriguing aspect pertains to the pronounced disparity in the sex composition of this cohort, with women constituting 77% of the participants. The longitudinal framework of this cohort offers a unique vantage point for monitoring health dynamics and disease progression over time. Among the 5,011 individuals drawn from the personnel of Siriraj Hospital, we achieved a remarkable 80.5% follow-up completion rate. Our findings illuminate the

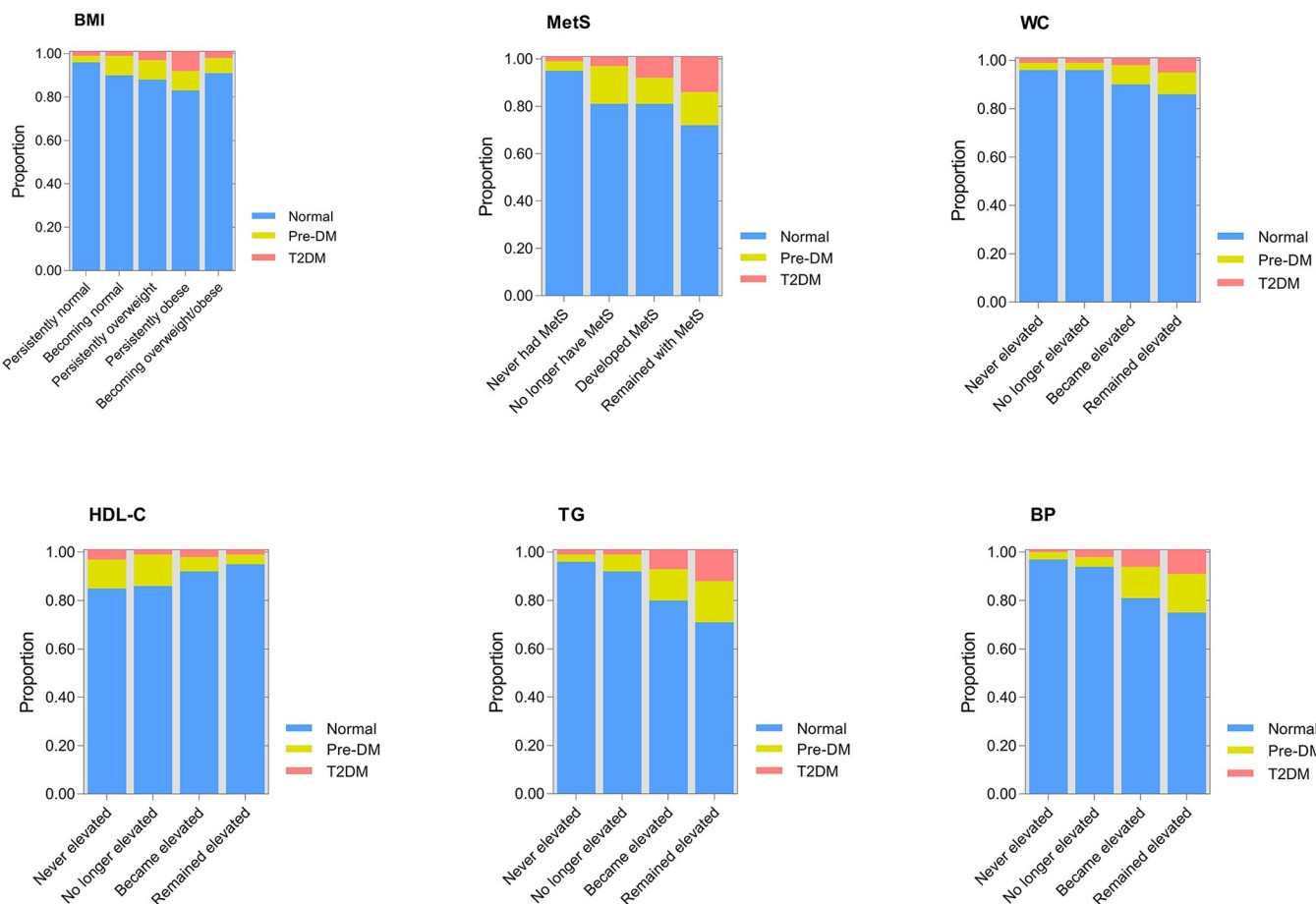

**Fig 3. Average predicted probabilities of remaining with normal blood glucose levels, developing pre-diabetes (Pre-DM), or developing type 2 diabetes (T2DM) in follow-up among participants with normal blood glucose levels (n = 2,477) at baseline.** BMI, body mass index; MetS, metabolic syndrome; WC, waist circumference; HDL-C, high-density lipoprotein cholesterol; TG, triglycerides; BP, blood pressure.

**Table 5. Genotypic variants of type 2 diabetes related single nucleotide polymorphisms association comparing pre-diabetes and type 2 diabetes mellitus to those without diabetes (control).**

| SNPs | Risk allele | % of Genotype | | | Risk factors for Pre-DM[a] | | Risk factors for T2DM[b] | |
|---|---|---|---|---|---|---|---|---|
| | | Control (n = 2,671) | Pre-DM[a] (n = 844) | T2DM[b] (n = 217) | OR (95%CI) | P-value[c] | OR (95%CI) | P-value[c] |
| rs7903146 | TT | 0.30 | 0.47 | 0.46 | 2.74 (0.32–23.3) | 0.411 | 1.05 (0.12–9.51) | 0.972 |
| rs12255372 | GT + TT | 6.36 | 6.64 | 8.76 | 1.33 (0.77–2.30) | 0.317 | 1.29 (0.77–2.17) | 0.342 |
| rs7917983 | TT | 16.74 | 15.17 | 14.75 | 0.84 (0.54–1.30) | 0.307 | 0.96 (0.61–1.52) | 0.804 |
| rs4506565 | TT | 0.30 | 0.47 | 0.46 | 2.71 (0.32–23.07) | 0.411 | 1.04 (0.11–9.47) | 0.971 |
| rs4132670 | GT + TT | 6.93 | 6.87 | 8.29 | 1.14 (0.67–1.93) | 0.629 | 1.18 (0.67–2.08) | 0.560 |
| rs12243326 | GT + TT | 6.36 | 6.40 | 8.29 | 1.24 (0.73–2.10) | 0.438 | 1.30 (0.74–2.28) | 0.366 |
| rs290487 | TT | 21.30 | 23.34 | 22.12 | 1.18 (0.78–1.79) | 0.715 | 0.96 (0.62–1.47) | 0.673 |

[a]Pre-DM was defined as a HbA1c level between 5.7 and 6.4%.

[b]T2DM was defined as a self-reported medical history of diabetes, current use of hypoglycemic medication, and/or a HbA1c level > 6.4%.

[c]P-value adjusted for age and sex; OR was an odd ratio with 95% confidence intervals (CI).

SNPs, single-nucleotide polymorphisms; pre-DM, pre-diabetes; T2DM, type 2 diabetes mellitus

prevalence of critical metabolic risk factors within this working population, including obesity, abdominal obesity, HTN, T2DM, DLP, and MetS. These revelations underscore the pressing health challenges confronting this cohort. Notably, the prevalence of T2DM and MetS exhibited an upsurge with advancing age and increasing BMI, reflecting a compelling association with both the aging demographic and the escalating obesity epidemic. This heightened prevalence carries significant implications for individual well-being and places an onus on healthcare systems, emphasizing the urgent need for preventive strategies and health promotion efforts.

In our study, we displayed a T2DM prevalence of 6% among the 5,011 participants, a figure that closely aligns with the 6th Thai National Health Examination Survey (NHES VI), which reported a 5.7% prevalence of diabetes among Thais aged 30–44 years [30]. Moreover, the observed T2DM prevalence in our study is similar to the 3.8% prevalence documented among 2,790 university hospital employees in Bangkok [31] and the 2.7% prevalence reported among 3,360 employees of the EGAT in Nonthaburi, an adjacent province of Bangkok [32]. The other striking revelation from this study was that 21% and 1.4% of participants were in unaware pre-DM and T2DM conditions, respectively. These findings indicated a significant association between T2DM and factors such as being overweight, obesity, smoking, and alcohol consumption, consistent with observations in heterogenous populations [33, 34]. Furthermore, HTN and DLP emerged as common comorbidities in the T2DM cohort, aligning with observations in diverse ethnic populations such as India and Saudi Arabia [35, 36].

Our multivariate analysis revealed several pivotal risk factors associated with T2DM. Age, MetS, HTN, and DLP prominently featured in this array. Notably, the strength of association for these predictor variables varied depending on diabetes status. For instance, while being elevated TG levels, HTN, DLP, and smoking were significantly associated with T2DM, they did not manifest such associations with pre-DM. The observed increasing odds of diabetes with age concurs with findings from other studies [37, 38], attributed to elevated glycated hemoglobin levels and changes in insulin sensitivity [39, 40]. Identifying adults with pre-DM assumes paramount importance for initiating early preventive or treatment measures, thereby mitigating the burden of diabetes, and reducing healthcare expenditures. In contrast to previous research indicating that women had a lower diabetes risk than men [41], our study suggests that women exhibited a heightened risk of pre-DM than men, potentially due to a higher proportion of women in the cohort leading to increased detection rates.

While our study aimed to explore the interplay between T2DM and genetic variants of the TCF7L2 gene, the results indicated no substantive evidence to substantiate a significant association between the analyzed variants and an augmented risk of T2DM within this cohort. Nevertheless, we acknowledge that other risk factors may play an influential role in the development of T2DM, thereby underscoring the complexity of the disease and the need for multifaceted investigations. Genotypic distribution analysis showed minimal variability of glycemic control between CC (wild type) and TT (mutant) of rs7917983 and rs290487 across all groups. The mutant genotype was found in all groups, except for rs12255372, rs4132670, and rs12243326 in the T2DM group, an observation that may be ascribed to the inclusion of younger participants in our study. Our findings align with prior studies that found no significant association of the SNPs with T2DM [42, 43]. Similar results were obtained in studies where the T allele had no impact on the association with T2DM [44, 45]. In contrast to our study, others concluded that the presence of rs7903146 C/T SNP was associated with an increased T2DM risk, particularly the homozygous TT genotype [46, 47]. Although no statistically significant difference emerged between the T2DM and non-DM groups regarding genotype or allele frequencies, the calculation of OR indicated that the genotypic distributions of the TCF7L2 rs7903146 (TT) and rs4506565 (TT) polymorphisms carried a risk for pre-DM, with an OR (95% CI) of 2.74 (0.32–23.3) and 2.71 (0.32–23.07), respectively.

The strengths of the SIH study encompass its urbanized, population-based design, the availability of repeated measures and biobank specimen storage, and annual follow-up conducted by healthcare specialists to ensure data precision and the punctual sample collection. SIH also recruits a diverse cross-section of individuals characterized by varying educational backgrounds and occupational profiles. This offers an invaluable resource for conducting workplace surveys to monitor transformations in risk determinants and explore causal relationships through interventional trials within a longitudinal cohort. However, our study has some limitations, which warrant consideration. It focuses on hospital personnel, predominantly women, which may restrict the generalizability of findings to the broader population and urban communities. Moreover, loss to follow-up and missing data could introduce bias. Maintaining the cohort and maximizing participant follow-up necessitate substantial efforts. Additionally, data on certain important potential predictors of diseases, such as associations between specific dietary patterns or eating behaviors and the outcome of interest, were not explored. Future research should delve deeper into the interplay between genetics, lifestyle factors, and T2DM development, with larger sample sizes and more diverse populations to validate findings and uncover potential genetic markers specific to this population.

## Conclusions

The SIH study provides valuable insights into the landscape of metabolic risk factors, highlighting concerning prevalence rates of conditions like obesity, abdominal obesity, HTN, T2DM, DLP, and MetS. These findings emphasize significant healthcare challenges, particularly in the context of an aging population and a growing obesity epidemic. The study reveals a 6% prevalence of T2DM, a figure that is in alignment with national and regional epidemiological data. Importantly, a substantial portion of the population remains unaware of their diabetes status, underscoring the need for proactive health promotion. Key contributors to onset of T2DM encompass advancing age, the presence of MetS, HTN, DLP, and the increases in TG concentrations as notable diabetes risk markers. While intended to explore T2DM's genetic basis, the analysis did not yield significant associations with the TCF7L2 gene, highlighting T2DM's multifaceted nature and its etiological underpinnings. Future research should incorporate larger, diverse cohorts to delve deeper into genetics, lifestyle, and T2DM, enhancing findings' validation and revealing population-specific genetic markers.

## Supporting information

**S1 Table. Methods for measuring laboratory parameters.**
(DOCX)

**S1 File. Genotyping analysis.**
(DOCX)

**S2 File.**
(DOCX)

**S1 Fig. Diabetes classification by self-report and HbA1c among the participants (SIH1-2).**
(DOCX)

## Acknowledgments

This research project is a collaboration between the Siriraj Medical Research Center and the Department of Preventive Medicine which is responsible for the staff health screening program. The authors gratefully acknowledge Prof. Dr. Ruengpung Sutthent for her guidance for

initiation of this project. We thank the members of the SIH Study Group, SPHERE staff members, and research assistants. The voluntary participation of all participants is highly appreciated. We are also grateful to the management of Siriraj Medical Research Center for providing office space, a recruitment center, and a biobank.

Members of SIH study group: Winai Ratanasuwan, Keerati Charoencholvanich, Bhoom Suktitipat, Manop Pithukpakorn, Prapat Suriyaphol, Rungroj Krittayaphong, Prasert Auewarakul, Chalermchai Mitrpant, Boonrat Tassaneetritap, Mayuree Homsanit, and Naravat Poungvarin.

Members of SPHERE group: Sureeporn Pumeiam, Bonggochpass Pinsawas, Pichanun Mongkolsucharitkul, Apinya Surawit, Tanyaporn Pongkunakorn, Sophida Suta, Thamonwan Manosan, Suphawan Ophakas, and Korapat Mayurasakorn.

Members of Biobank: Somruedee Chatsiricharoenkul, Parichart Permpikul, Duangthip Apiratmontree, Sutee Udomchotphruet, and Pattranit Onsing.

## Author Contributions

**Conceptualization:** Sureeporn Pumeiam, Korapat Mayurasakorn.

**Data curation:** Apinya Surawit, Sureeporn Pumeiam.

**Formal analysis:** Apinya Surawit.

**Funding acquisition:** Winai Ratanasuwan, Mayuree Homsanit, Keerati Charoencholvanich, Yuthana Udomphorn, Bhoom Suktitipat, Korapat Mayurasakorn.

**Investigation:** Pichanun Mongkolsucharitkul, Apinya Surawit, Thamonwan Manosan, Suphawan Ophakas, Sophida Suta, Bonggochpass Pinsawas, Tanyaporn Pongkunakorn, Sureeporn Pumeiam.

**Methodology:** Pichanun Mongkolsucharitkul, Apinya Surawit, Thamonwan Manosan, Suphawan Ophakas, Sophida Suta, Bonggochpass Pinsawas, Tanyaporn Pongkunakorn, Sureeporn Pumeiam.

**Project administration:** Winai Ratanasuwan, Mayuree Homsanit, Keerati Charoencholvanich, Yuthana Udomphorn, Bhoom Suktitipat, Korapat Mayurasakorn.

**Writing – original draft:** Pichanun Mongkolsucharitkul, Sophida Suta, Korapat Mayurasakorn.

**Writing – review & editing:** Pichanun Mongkolsucharitkul, Apinya Surawit, Thamonwan Manosan, Suphawan Ophakas, Sophida Suta, Bonggochpass Pinsawas, Tanyaporn Pongkunakorn, Sureeporn Pumeiam, Winai Ratanasuwan, Mayuree Homsanit, Keerati Charoencholvanich, Yuthana Udomphorn, Bhoom Suktitipat, Korapat Mayurasakorn.

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
