## [Decision Letter · Decision Letter 0]

2 Jan 2024

PONE-D-23-40874Metabolic and genetic risk factors associated with pre-diabetes and type 2 diabetes in Thai healthcare employees: a long-term study from the Siriraj Health (SIH) Cohort StudyPLOS ONE

Dear Dr. Mayurasakorn,

Thank you for submitting your manuscript to PLOS ONE. After careful consideration, we feel that it has merit but does not fully meet PLOS ONE’s publication criteria as it currently stands. Therefore, we invite you to submit a revised version of the manuscript that addresses the points raised during the review process.

We look forward to receiving your revised manuscript.

Kind regards,

Chikezie Hart Onwukwe

Academic Editor

PLOS ONE

Journal Requirements:

2. For studies involving third-party data, we encourage authors to share any data specific to their analyses that they can legally distribute. PLOS recognizes, however, that authors may be using third-party data they do not have the rights to share. When third-party data cannot be publicly shared, authors must provide all information necessary for interested researchers to apply to gain access to the data. (https://journals.plos.org/plosone/s/data-availability#loc-acceptable-data-access-restrictions) 

3. One of the noted authors is a group or consortium "Siriraj Population Health and Nutrition Research (SPHERE) group and SIH study group". In addition to naming the author group, please list the individual authors and affiliations within this group in the acknowledgments section of your manuscript. Please also indicate clearly a lead author for this group along with a contact email address.’ 

4. We note that Figure 1 in your submission contain map/satellite images which may be copyrighted. All PLOS content is published under the Creative Commons Attribution License (CC BY 4.0), which means that the manuscript, images, and Supporting Information files will be freely available online, and any third party is permitted to access, download, copy, distribute, and use these materials in any way, even commercially, with proper attribution. For these reasons, we cannot publish previously copyrighted maps or satellite images created using proprietary data, such as Google software (Google Maps, Street View, and Earth). For more information, see our copyright guidelines: http://journals.plos.org/plosone/s/licenses-and-copyright.

Reviewers' comments:

Reviewer's Responses to Questions

**Comments to the Author**

1. Is the manuscript technically sound, and do the data support the conclusions?

Reviewer #1: Yes

Reviewer #2: Yes

Reviewer #3: Partly

2. Has the statistical analysis been performed appropriately and rigorously? 

Reviewer #1: Yes

Reviewer #2: Yes

Reviewer #3: I Don't Know

3. Have the authors made all data underlying the findings in their manuscript fully available?

Reviewer #1: No

Reviewer #2: Yes

Reviewer #3: Yes

4. Is the manuscript presented in an intelligible fashion and written in standard English?

Reviewer #1: Yes

Reviewer #2: No

Reviewer #3: Yes

5. Review Comments to the Author

Reviewer #1: There is no new massage in this study. However, there is severe concern about the cohort sample and it seems there is sampling bias with two incoherent . A large portion of sample (4,518 of 5011) aged 18-55 and a small portion (493 of 5011) aged more than 55 years and this affect the results. These two age groups have different characteristics including increased rate of diabetes and metabolic syndrome (which is the target of this study) in the second group but all of them have been analyzed with each other that biased the results. This bias also affects the result of genotyping study because many individuals in first age group (younger group) will develop diabetes in the next ages above 50 and belong to diabetes group but currently are in normal group.

The data should be re-analyzed after stratification of two samples and results should be arranged and presented after new analysis.

Reviewer #2: Dear Authors,

This manuscript addresses a very relevant subject, considering the increasing prevalence of type 2 diabetes and its complications, besides the evidence on the role of the TCF7L2 gene on the liability of developing type 2 diabetes related to some specific polymorphism, which may have differences among different populations that need clarifications. This manuscript deserves some grammar and punctuation revision. As pointed out by the authors, the lack of data related especially to HbA1C in the follow-up may have interfered with the final results of diabetes diagnosis. It reinforces the need for a more extensive and detailed study.

Yours Sincerely,

Reviewer #3: Pichanun Mongkolsucharitkul et al has submitted the research study “Metabolic and genetic risk factors associated with pre-diabetes and type 2 diabetes in Thai healthcare 2 employees: a long-term study from the Siriraj Health (SIH) Cohort Study” for review.

The study aims to explore NCD risk factors, biomarker relationships, and to develop a T2DM risk prediction model while 88 investigating the association between T2DM and genetic variants of the TCF7L2 gene.

My review comments on the manuscript as below:

1) The chosen research question is very much relevant to the ever-increasing prevalence of Diabetes in the world.

2) I request the author to proofread the manuscript for grammatic errors and language (e.g line 32, 33)

3) Methods:

a) Exclusion criteria and inclusion criteria may be defined more specifically.

b) S1 Fig: Why the ‘self-reported uncertain DM’ is excluded?

4) What are the criteria for metabolic syndrome, please define (line 198) how it has been coded as Yes and No

5) Research tools used, for example, the questionnaire used for participant interviews, Food frequency questionnaire is not submitted for review

6) Line 203- please explain the limitations of follow up data collection more explicitly. For example, if the participants were physically available or not,

7) Line 223 - T2DM with concurrent HT - data not given. Moreover, HT data could be given as systolic, diastolic range that is specifically coexist with DM.

8) Line 225 – Micro and macroalbuminuria, by itself are not modifiable. Only the progression to renal complications can be reduced with strict glycemic and HTN control. Please explain

9) Line 251 - 253: this observation may be explained more elaborately using the results of the data analysis.

10) Line 303 – repeated word- microalbuminuria

11) Genetic analysis was not reviewed

6. PLOS authors have the option to publish the peer review history of their article (what does this mean?). If published, this will include your full peer review and any attached files.

Reviewer #1: No

Reviewer #2: No

Reviewer #3: No

---

## [Author Response · Author response to Decision Letter 0]

18 Feb 2024

February 19, 2024

Dear Editor,

We thank you for the opportunity to re-submit our revised manuscript entitled “Metabolic and genetic risk factors associated with pre-diabetes and type 2 diabetes in Thai healthcare employees: a long-term study from the Siriraj Health (SIH) Cohort Study” (Manuscript ID: PONE-D-23-40874)

The reviewers’ comments were insightful. We have responded specifically to each suggestion below. To make changes easier to be identified, I have highlighted them in yellow in the text. 

The manuscript has been reviewed and approved by all of the authors. Data in this manuscript have not been previously reported/published nor is it being considered for publication elsewhere. All correspondence, editorial communications, and reprint requests should be sent to the above address.

Our responses to the editor and reviewers’ comments are below. 

We strongly hope you find our manuscript suitable for publication and look forward to hearing from you. 

Sincerely,

Associate Professor Korapat Mayurasakorn, MD, FRCFPT 

Family Physician

Lecturer

Email: korapat.may@mahidol.ac.th

Response to Editor

Point 1: Please ensure that your manuscript meets PLOS ONE's style requirements, including those for file naming. The PLOS ONE style templates can be found at

Response 1: Thank you for your comment. We will ensure that our manuscript meets PLOS ONE's style requirements.

Point 2: For studies involving third-party data, we encourage authors to share any data specific to their analyses that they can legally distribute. PLOS recognizes, however, that authors may be using third-party data they do not have the rights to share. When third-party data cannot be publicly shared, authors must provide all information necessary for interested researchers to apply to gain access to the data. (https://journals.plos.org/plosone/s/data-availability#loc-acceptable-data-access-restrictions) 

Response 2: Thank you for your comment. We have addressed the data availability statement as follows. The study is a part of the Siriraj Health (SIH) Cohort Study. The authors do not have the legal authority to distribute the data. However, all interested researchers can follow the respective rules and protocols of data sharing and scientific collaboration, just as the current authors have. The data are accessible upon request from the Siriraj Health study committee via email: sihealthstudy@mahidol.edu.

Point 3: One of the noted authors is a group or consortium "Siriraj Population Health and Nutrition Research (SPHERE) group and SIH study group". In addition to naming the author group, please list the individual authors and affiliations within this group in the acknowledgments section of your manuscript. Please also indicate clearly a lead author for this group along with a contact email address.’ 

Response 3: Thank you for your comment. We have decided to remove the “Siriraj Population Health and Nutrition Research (SPHERE) group and SIH study group" as authors, as most of members of these groups are already included as authors, and the SPHERE group is also listed in the author’s affiliations. Instead, we will acknowledge the individual members of these groups. 

Point 4: We note that Figure 1 in your submission contain map/satellite images which may be copyrighted. All PLOS content is published under the Creative Commons Attribution License (CC BY 4.0), which means that the manuscript, images, and Supporting Information files will be freely available online, and any third party is permitted to access, download, copy, distribute, and use these materials in any way, even commercially, with proper attribution. For these reasons, we cannot publish previously copyrighted maps or satellite images created using proprietary data, such as Google software (Google Maps, Street View, and Earth). For more information, see our copyright guidelines: http://journals.plos.org/plosone/s/licenses-and-copyright.

“I request permission for the open-access journal PLOS ONE to publish XXX under the Creative Commons Attribution License (CCAL) CC BY 4.0

(http://creativecommons.org/licenses/by/4.0/). Please be aware that this license allows unrestricted use and distribution, even commercially, by third parties. Please reply and provide explicit written permission to publish XXX under a CC BY license and complete the attached form.”

While revising your submission, please upload your figure files to the Preflight Analysis and Conversion Engine (PACE) digital diagnostic tool,

https://pacev2.apexcovantage.com/. PACE helps ensure that figures meet PLOS requirements. To use PACE, you must first register as a user. Registration is free. Then, login and navigate to the UPLOAD tab, where you will find detailed instructions on how to use the tool. If you encounter any issues or have any questions when using PACE, please email PLOS at figures@plos.org. Please note that Supporting Information files do not need this step.

Response 4: Thank you for your concern. The map images in Figure 1 were created by Apinya Surawit, who is one of the authors, to use as the original images for illustrative purposes in our study only. We also uploaded the illustration file (final5.ai) that was the original file of the map images as the attached file in the other document. All of the figure files were already uploaded to the Preflight Analysis and Conversion Engine (PACE) digital diagnostic tool. And Ms. Apinya Surawit has provided a signed content permission form.

 

Response to Reviewers

Reviewer #1

Point 1: There is no new massage in this study. However, there is severe concern about the cohort sample and it seems there is sampling bias with two incoherent. A large portion of sample (4,518 of 5011) aged 18-55 and a small portion (493 of 5011) aged more than 55 years and this affect the results. These two age groups have different characteristics including increased rate of diabetes and metabolic syndrome (which is the target of this study) in the second group but all of them have been analyzed with each other that biased the results. This bias also affects the result of genotyping study because many individuals in first age group (younger group) will develop diabetes in the next ages above 50 and belong to diabetes group but currently are in normal group. The data should be re-analyzed after stratification of two samples and results should be arranged and presented after new analysis.

Response 1: Thank you for your feedback. We have carefully reviewed your concerns regarding potential sampling bias in our study cohort and acknowledge the importance of addressing this issue to ensure the validity of our results. We also would like to clarify that in our first enrollment, known as "Siriraj Health Study 1 (SIH-1)," we had 4,528 participants aged 18-55 years, while the second enrollment, which began in January 2020, consisted of 493 participants aged over 18 years old.

We have rephrased in Line 108-110, “Beginning in January 2020, the second phase of the Siriraj Health (SIH2) study included participants aged over 18 years with no upper age limit. However, due to the COVID-19 pandemic from 2020 to 2022, only 493 participants were enrolled.” It is worth noting that participants in both SIH1 and SIH2 did not exhibit significantly different age-group distributions, as illustrated in the following unpublished table (for your information only).

Age, years SIH1 (n=4,518) SIH2 (n=493) Total (n=5,011)

 n % n % n %

< 30 1,258 27.84 157 31.85 1,415 28.24

30 - 39 1,899 42.03 174 35.29 2,073 41.37

40 - 49 1,038 22.97 91 18.46 1,129 22.53

50 - 55 322 7.13 28 5.68 350 6.98

≥ 56 1 0.02 43 8.72 44 0.88

We also reanalyzed the progression of diabetes development among individuals who were not diagnosed with diabetes to evaluate the relative risk of developing pre-diabetes or diabetes at follow-up as shown in Table 4 and Fig 2.

 

Reviewer #2

Point 1: This manuscript addresses a very relevant subject, considering the increasing prevalence of type 2 diabetes and its complications, besides the evidence on the role of the TCF7L2 gene on the liability of developing type 2 diabetes related to some specific polymorphism, which may have differences among different populations that need clarifications. This manuscript deserves some grammar and punctuation revision. As pointed out by the authors, the lack of data related especially to HbA1C in the follow-up may have interfered with the final results of diabetes diagnosis. It reinforces the need for a more extensive and detailed study.

Response 1: Thank you for your feedback. We appreciate your acknowledgment of the relevance of our study topic and the importance of addressing potential genetic influences on type 2 diabetes (T2DM) susceptibility, particularly related to the TCF7L2 gene. We have explained more about TCF7L2 gene in Line 86-90, “In Asian populations, including Japanese [22], Thai [23], and Chinese [24], variants of the TCF7L2 gene, such as rs7903146, rs11196205, and rs12255372, have been identified as significant genetic risk factors for T2DM. These findings highlight the genetic heterogeneity of T2DM across different ethnic groups and underscore the importance of understanding population-specific genetic determinants of the disease.”

We acknowledge your suggestion for grammar and punctuation revision, and we will ensure to thoroughly review and enhance these aspects to improve the clarity and readability of the manuscript. Additionally, we acknowledge the importance of addressing the highlighted limitations regarding the lack of data related to HbA1C in the follow-up period due to the nature of a routine annual health examination policy, which may have influenced the final results of diabetes diagnosis, as indicated in Line 224-228, 423-424. However, we would like to clarify that although we lack HbA1C data, we do possess information related to fasting blood glucose collected during the follow-up period. This enables us to monitor the progression of diabetes development among individuals who were not diagnosed with diabetes at baseline, as illustrated in Table 4 and Fig 2.

 

Reviewer #3

Point 1: Pichanun Mongkolsucharitkul et al has submitted the research study.

“Metabolic and genetic risk factors associated with pre-diabetes and type 2 diabetes in Thai healthcare 2 employees: a long-term study from the Siriraj Health (SIH) Cohort Study” for review. The study aims to explore NCD risk factors, biomarker relationships, and to develop a T2DM risk prediction model while investigating the association between T2DM and genetic variants of the TCF7L2 gene.

My review comments on the manuscript as below:

1) The chosen research question is very much relevant to the ever-increasing prevalence of Diabetes in the world.

Response 1: Thank you for recognizing the relevance of our chosen research question. We agree that the prevalence of diabetes globally is a critical issue, and we believe that addressing this research question will contribute to a better understanding of the challenges and potential solutions in managing this health concern. We are committed to conducting thorough and insightful research that can offer valuable insights into addressing the complexities surrounding diabetes. And we will continue to monitor the progress of NCD for this cohort.

Point 2: I request the author to proofread the manuscript for grammatic errors and language (e.g line 32, 33)

Response 2: Thank you for your comment. We acknowledge your suggestion for grammar and language revision, and we will ensure to thoroughly review and enhance these aspects to improve the clarity and readability of the manuscript (highlighted in yellow).

Point 3: Methods: Exclusion criteria and inclusion criteria may be defined more specifically.

Response 3: Thank you for your comment. We have added the inclusion and exclusion criteria in Line 100-104, “The inclusion criteria of the study were the Siriraj Hospital personnel who attended an annual health screening surveillance program. The exclusion criteria of the study were individuals who withheld treatment information, unable to consistently follow-up, unable to participate in this cohort in the next 2 years, or presence of contraindications for blood sampling, such as blood clotting.”

Point 4: S1 Fig: Why the ‘self-reported uncertain DM’ is excluded?

Response 4: Thank you for your comment. The exclusion of individuals who self-reported uncertainty about whether they have diabetes, along with those with other missing data, was a decision made due to the limitations inherent in the survey data collection process. When participants are unsure about their diabetes status or when there are missing data points, it can introduce ambiguity and potential inaccuracies into the analysis. To maintain the reliability and validity of the study results, the research team opted to exclude these individuals from the analysis in this instance.

Point 5: What are the criteria for metabolic syndrome, please define (line 198) how it has been coded as Yes and No

Response 5: Thank you for your comment. We have added the criteria for metabolic syndrome in Line 158-162, “We used the criteria of metabolic syndrome (MetS) in adults by the IDF definition with South Asian e

---

## [Decision Letter · Decision Letter 1]

10 Mar 2024

PONE-D-23-40874R1Metabolic and genetic risk factors associated with pre-diabetes and type 2 diabetes in Thai healthcare employees: a long-term study from the Siriraj Health (SIH) Cohort StudyPLOS ONE

Dear Dr. Mayurasakorn,

Thank you for submitting your manuscript to PLOS ONE. After careful consideration, we feel that it has merit but does not fully meet PLOS ONE’s publication criteria as it currently stands. Therefore, we invite you to submit a revised version of the manuscript that addresses the points raised during the review process.

We look forward to receiving your revised manuscript.

Kind regards,

Chikezie Hart Onwukwe

Academic Editor

PLOS ONE

Journal Requirements:

Reviewers' comments:

Reviewer's Responses to Questions

**Comments to the Author**

1. If the authors have adequately addressed your comments raised in a previous round of review and you feel that this manuscript is now acceptable for publication, you may indicate that here to bypass the “Comments to the Author” section, enter your conflict of interest statement in the “Confidential to Editor” section, and submit your "Accept" recommendation.

Reviewer #2: All comments have been addressed

2. Is the manuscript technically sound, and do the data support the conclusions?

Reviewer #2: Partly

3. Has the statistical analysis been performed appropriately and rigorously? 

Reviewer #2: Yes

4. Have the authors made all data underlying the findings in their manuscript fully available?

Reviewer #2: (No Response)

5. Is the manuscript presented in an intelligible fashion and written in standard English?

Reviewer #2: Yes

6. Review Comments to the Author

Reviewer #2: Dear Authors,

Considering Siriraj Hospital personnel are involved in an annual health screening surveillance program, and about half the population is overweight and obese, it is expected that they receive diet and exercise orientations. As you didn't check behavioral changes, fasting blood glucose may not be enough to diagnose all participants and mislead the conclusions.

You spent much of your manuscript discussing the well-known risk factors for developing diabetes. I expected some comments on the possible influence of such metabolic factors on the outcomes you may have found in your genetic findings and comparisons of different age groups.

7. PLOS authors have the option to publish the peer review history of their article (what does this mean?). If published, this will include your full peer review and any attached files.

Reviewer #2: No

---

## [Author Response · Author response to Decision Letter 1]

27 Mar 2024

Dear Editor,

We thank you for the opportunity to re-submit our revised manuscript entitled “Metabolic and genetic risk factors associated with pre-diabetes and type 2 diabetes in Thai healthcare employees: a long-term study from the Siriraj Health (SIH) Cohort Study” (Manuscript ID: PONE-D-23-40874R1)

The reviewers’ comments were insightful. We have responded specifically to each suggestion below. To make changes easier to be identified, I have highlighted them in yellow in the text. 

The manuscript has been reviewed and approved by all of the authors. Data in this manuscript have not been previously reported/published nor is it being considered for publication elsewhere. All correspondence, editorial communications, and reprint requests should be sent to the above address.

Our responses to the editor and reviewers’ comments are below. 

We strongly hope you find our manuscript suitable for publication and look forward to hearing from you. 

Sincerely,

Associate Professor Korapat Mayurasakorn, MD, FRCFPT 

Family Physician

Lecturer

Email: korapat.may@mahidol.ac.th

Response to Editor

Point 1: Please review your reference list to ensure that it is complete and correct. If you have cited papers that have been retracted, please include the rationale for doing so in the manuscript text or remove these references and replace them with relevant current references. Any changes to the reference list should be mentioned in the rebuttal letter that accompanies your revised manuscript. If you need to cite a retracted article, indicate the article’s retracted status in the References list and also include a citation and full reference for the retraction notice.

Response 1: Thank you for your comment. We acknowledge your suggestion for reference revision, and we will ensure to thoroughly recheck and enhance these aspects to improve the correct template of the reference list (highlighted in yellow). We did not cite any retracted articles in this manuscript.

Response to Reviewers

Reviewer #2

Point 1: Dear Authors,

Considering Siriraj Hospital personnel are involved in an annual health screening surveillance program, and about half the population is overweight and obese, it is expected that they receive diet and exercise orientations. As you didn't check behavioral changes, fasting blood glucose may not be enough to diagnose all participants and mislead the conclusions.

You spent much of your manuscript discussing the well-known risk factors for developing diabetes. I expected some comments on the possible influence of such metabolic factors on the outcomes you may have found in your genetic findings and comparisons of different age groups.

Response 1: Thank you for your insightful comments and suggestions regarding our manuscript. We appreciate your observation regarding the importance of assessing behavioral changes alongside metabolic markers in our study population. We agree that incorporating such assessments would provide a more comprehensive understanding of the outcomes.

Regarding the involvement of Siriraj Hospital personnel in an annual health screening surveillance program, it's pertinent to note that while the hospital does provide health check-up reports to its workers, there's no mandatory provision of dietary and exercise programs. Consequently, behavioral changes primarily occur on a voluntary basis. We will certainly consider including measures of behavioral changes in future studies to address this limitation and provide a more robust analysis.

We acknowledge that relying solely on fasting blood glucose (FBG) levels may not capture all participants at risk for diabetes. In Thailand, FBG levels serve as a standard early health screening measure, offering insights into glucose trends over time. While we did not utilize HbA1c in screening, recognizing its higher accuracy in defining diabetes, implementing more complicated tests could have resulted in increased missing data due to the logistical challenges associated with obtaining blood samples from the population.

However, it's essential to acknowledge the fundamental differences in the pathophysiology between HbA1c and fasting blood glucose. HbA1c reflects the average blood glucose levels over the past 2-3 months, providing a more stable indicator of long-term glycemic control, whereas fasting blood glucose offers a snapshot of blood sugar levels at a specific point in time. Despite these differences, in our pursuit of precise predictions, we attempted to compare HbA1c values with a limited number of samples, contrasting them with the FBG values that were tracked. Although the pathophysiology underlying HbA1c and FBG levels may differ, it was observed that the trends of both values move in the same direction, as illustrated in the following unpublished figure (for your information only).

Furthermore, we understand your expectation for a discussion on the influence of metabolic factors on our genetic findings and comparisons across different age groups. Indeed, the observed association between the TCF7L2 gene (all 7 SNPs) and the decreased risk of type 2 diabetes mellitus (T2DM) in individuals aged 40 and above could potentially be influenced by statistical fluctuations, especially in studies with small sample sizes. This concern is particularly relevant when studying complex traits like T2DM, where genetic associations may be subtle and influenced by various environmental and genetic factors. Additionally, the researchers did not conduct a subgroup study to analyze genetic risk factors in relation to other metabolic risk factors, given the small size of the genetic sample.

As illustrated in the following unpublished table (for your information only), we observed significant associations between TCF7L2 SNPs and both pre-DM and T2DM across different age strata. Notably, the rs12243326, rs12255372, rs290487, rs4132670, rs4506565, rs7903146, and rs7917983 variants exhibited varying degrees of association with pre-DM and T2DM risk, with some associations being more pronounced in certain age groups.

Regarding the influence of metabolic factors on our genetic findings, we conducted adjusted analyses, accounting for sex, body mass index (BMI), waist circumference (WC), total cholesterol (TC), triglycerides (TG), high-density lipoprotein (HDL), and low-density lipoprotein (LDL). These adjustments allowed us to assess the independent effects of TCF7L2 SNPs on pre-DM and T2DM risk while controlling for potential confounders related to metabolic health.

---

## [Decision Letter · Decision Letter 2]

19 Apr 2024

Metabolic and genetic risk factors associated with pre-diabetes and type 2 diabetes in Thai healthcare employees: a long-term study from the Siriraj Health (SIH) Cohort Study

PONE-D-23-40874R2

Dear Dr. Korapat Mayurasakorn,

We’re pleased to inform you that your manuscript has been judged scientifically suitable for publication and will be formally accepted for publication once it meets all outstanding technical requirements.

Kind regards,

Chikezie Hart Onwukwe

Academic Editor

PLOS ONE

Additional Editor Comments (optional):

Reviewers' comments:

Reviewer's Responses to Questions

**Comments to the Author**

1. If the authors have adequately addressed your comments raised in a previous round of review and you feel that this manuscript is now acceptable for publication, you may indicate that here to bypass the “Comments to the Author” section, enter your conflict of interest statement in the “Confidential to Editor” section, and submit your "Accept" recommendation.

Reviewer #2: All comments have been addressed

2. Is the manuscript technically sound, and do the data support the conclusions?

Reviewer #2: Yes

3. Has the statistical analysis been performed appropriately and rigorously? 

Reviewer #2: Yes

4. Have the authors made all data underlying the findings in their manuscript fully available?

Reviewer #2: Yes

5. Is the manuscript presented in an intelligible fashion and written in standard English?

Reviewer #2: Yes

6. Review Comments to the Author

Reviewer #2: (No Response)

7. PLOS authors have the option to publish the peer review history of their article (what does this mean?). If published, this will include your full peer review and any attached files.

Reviewer #2: No

---

## [Editor Report · Acceptance letter]

26 Apr 2024

PONE-D-23-40874R2 

PLOS ONE

Dear Dr. Mayurasakorn, 

I'm pleased to inform you that your manuscript has been deemed suitable for publication in PLOS ONE. Congratulations! Your manuscript is now being handed over to our production team.

Kind regards, 

on behalf of

Dr. Chikezie Hart Onwukwe 

Academic Editor

PLOS ONE